# An integrated multi-omics analysis identifies prognostic molecular subtypes of non-muscle-invasive bladder cancer

Sia Viborg Lindskrog [1,2,29], Frederik Prip[1,2,29], Philippe Lamy [1], Ann Taber [1,2], Clarice S. Groeneveld [3,4], Karin Birkenkamp-Demtröder [1,2], Jørgen Bjerggaard Jensen[2,5], Trine Strandgaard[1,2], Iver Nordentoft [1], Emil Christensen [1,2], Mateo Sokac[1,2], Nicolai J. Birkbak [1,2], Lasse Maretty[1,2], Gregers G. Hermann[6], Astrid C. Petersen[7], Veronika Weyerer[8], Marc-Oliver Grimm[9], Marcus Horstmann[9,10], Gottfrid Sjödahl [11], Mattias Höglund[12], Torben Steiniche[13], Karin Mogensen[6], Aurélien de Reyniès[3], Roman Nawroth[14], Brian Jordan[15], Xiaoqi Lin[15], Dejan Dragicevic[16], Douglas G. Ward[17], Anshita Goel[17], Carolyn D. Hurst[18], Jay D. Raman[19], Joshua I. Warrick [20], Ulrika Segersten[21], Danijel Sikic[22], Kim E. M. van Kessel[23], Tobias Maurer[14,24], Joshua J. Meeks[15], David J. DeGraff[20], Richard T. Bryan[17], Margaret A. Knowles[18], Tatjana Simic[25], Arndt Hartmann[8], Ellen C. Zwarthoff[23], Per-Uno Malmström[21], Núria Malats [26], Francisco X. Real [27,28] & Lars Dyrskjøt [1,2✉]

The molecular landscape in non-muscle-invasive bladder cancer (NMIBC) is characterized by large biological heterogeneity with variable clinical outcomes. Here, we perform an integrative multi-omics analysis of patients diagnosed with NMIBC ($n = 834$). Transcriptomic analysis identifies four classes (1, 2a, 2b and 3) reflecting tumor biology and disease aggressiveness. Both transcriptome-based subtyping and the level of chromosomal instability provide independent prognostic value beyond established prognostic clinicopathological parameters. High chromosomal instability, p53-pathway disruption and APOBEC-related mutations are significantly associated with transcriptomic class 2a and poor outcome. RNA-derived immune cell infiltration is associated with chromosomally unstable tumors and enriched in class 2b. Spatial proteomics analysis confirms the higher infiltration of class 2b tumors and demonstrates an association between higher immune cell infiltration and lower recurrence rates. Finally, the independent prognostic value of the transcriptomic classes is documented in 1228 validation samples using a single sample classification tool. The classifier provides a framework for biomarker discovery and for optimizing treatment and surveillance in next-generation clinical trials.

A full list of author affiliations appears at the end of the paper.

Urothelial non-muscle-invasive bladder cancer (NMIBC) represents the most common type of bladder cancer. Patients with NMIBC experience a high likelihood of disease recurrence (50–70%) and progression to muscle-invasive bladder cancer (MIBC; up to 20%, depending on stage and grade)[1]. Consequently, although 5-year survival rates are favorable (>90%), most patients must undergo lifelong cystoscopic surveillance and multiple therapeutic interventions, making bladder cancer the most expensive cancer to treat[2]. Clinically, high-risk NMIBC is treated with adjuvant intravesical instillations of Bacillus Calmette–Guérin (BCG) after surgery to eradicate residual disease and hence reduce the frequency of recurrence and progression[1].

Despite similar clinical and histopathological characteristics, tumors show large differences in disease aggressiveness and response to therapy, emphasizing the urgent need for further delineation of clinically useful biomarker tests to facilitate and improve patient surveillance and treatment[3]. Earlier studies of NMIBC biology addressed gene expression for classification of aggressiveness, resulting in the identification of two major molecular subtypes[4–6]. When considering the whole spectrum of disease stages, five subtypes of bladder cancer were identified; in particular, three subtypes (Urothelial-like, genomically unstable, and a group of infiltrated cases) were associated with NMIBC[7]. In a more recent study of 460 NMIBC patients, we reported three gene expression-based classes (class 1–3; UROMOL2016 classification system) with different clinical outcomes and molecular characteristics[8]. Differences in biological processes, such as cell cycle, epithelial-mesenchymal transition (EMT), and differentiation, were observed. Furthermore, mutations in well-known cancer driver genes, i.e., TP53 and ERBB2, were primarily found in high-risk class 2 tumors, together with enrichment for APOBEC-related mutational processes.

Analysis of genomic alterations in NMIBC has revealed complex genomic patterns underlying bladder carcinogenesis. Activating mutations in FGFR3 and PIK3CA and chromosome 9 deletions have been identified as early disease drivers[9–11]. Recently, van Kessel et al. showed that NMIBC at high risk for progression could be further subdivided into good, moderate, and poor progression risk groups based on mutations in FGFR3 and methylation of GATA2[12]. Hurst et al. assessed 160 tumors for genome-wide copy number alterations (CNAs) using array-based comparative genomic hybridization (CGH). The study included 49 high-grade T1 tumors that separated into three major genomic subgroups, one of which contained the majority of tumors showing disease progression[13]. In a more recent study, the same group analyzed CNAs in 140 Ta tumors and identified two major genomic subtypes (GS1 and GS2). GS1 tumors showed no or very few CNAs, while tumors in GS2 showed more alterations and a high frequency of chromosome 9 deletions[14]. Exome sequencing of 28 Ta tumors revealed that GS2 tumors had a higher mutational load with enrichment for APOBEC-related mutations compared to GS1 tumors. Furthermore, comparing 79 of the samples to transcriptional subtypes showed that the tumors were primarily classified as the Urothelial-like A subtype (Lund Taxonomy). Application of the UROMOL2016 classification system showed that GS2 tumors with higher genomic instability were enriched for the class 2 subtype[14]. However, additional refinement of these genomic studies is required to determine optimal predictors of disease aggressiveness and outcome.

The tumor microenvironment has also been linked to prognosis in NMIBC. A high infiltration of cytotoxic T lymphocytes (CTLs) is associated with better prognosis in many cancer types, including MIBC[15,16]. In contrast, high infiltration of tumor infiltrating-lymphocytes (TILs) has been associated with progression in NMIBC[17,18]. Furthermore, the presence of tumor-associated macrophages and mature tumor-infiltrating dendritic cells has been related to progression of NMIBC[19]. The impact of regulatory T cells (Tregs) is conflicting, since high levels of Treg infiltration has been associated with both a favorable[20] and unfavorable prognosis of bladder cancer[21,22]. The impact of immune cell infiltration on disease outcome and association with molecular subtypes and genomic alterations in NMIBC needs to be further studied.

Overall, our understanding of the molecular landscape of NMIBC is still incomplete, and integrative multi-omics analysis is needed to obtain further knowledge of biological processes contributing to disease aggressiveness, recurrence, and progression. This should ultimately lead to biomarker-based optimized surveillance and therapy modalities for patients with NMIBC.

Here, we report an integrative multi-omics analysis of NMIBC tumors from a total of 834 patients included in the UROMOL project. With this analysis, we delineate genomic and transcriptomic predictors of outcome in NMIBC, and present an online tool for the classification of independent samples (http://nmibc-class.dk).

## Results

**Clinical, pathological, and molecular information.** Patients were enrolled in the UROMOL project, a European multicenter prospective study of NMIBC. The initial reports from the UROMOL project included only transcriptomic analysis[6,8]. We have now performed an integrated multi-omics analysis and have expanded the work to a larger NMIBC patient series with updated follow-up that is essential to acquire insight into the implications for patient management. In total, 862 tumors (613 Ta, 238 T1, 11 carcinoma in situ (CIS)) were analyzed in this study. Median follow-up for patients without progression was 49 months and 10.3% progressed to MIBC. A detailed summary of clinical and histopathological information and the analyses performed is provided in Supplementary Table 1.

**Delineation of transcriptomic classes in NMIBC.** We analyzed bulk RNA-Sequencing (RNA-Seq) data from 535 patients (397 Ta, 135 T1, 3 CIS; an expansion of the 460 NMIBC patient cohort previously analyzed[8]). Using unsupervised consensus clustering of gene-based expression values restricted to the 4000 genes with highest variation across the dataset[23], we identified four transcriptomic classes which partially overlapped with the previous UROMOL2016 classes 1–3: high-risk class 2 was further subdivided into two subclasses, named class 2a and 2b for continuity (Fig. 1a, b). Classes showed significantly different progression-free survival (PFS, $p = 6.6 \times 10^{-5}$; log-rank test; Fig. 1c): patients with class 2a tumors had the worst outcome, followed by patients with class 2b tumors. Patients with class 1 tumors had the best recurrence-free survival (RFS, $p = 0.025$; log-rank test; Fig. 1d). Multivariable Cox regression analysis revealed that high-risk class 2a and 2b were independently associated with worse PFS and RFS when adjusted for the clinical EORTC risk score (European Organisation for Research and Treatment of Cancer[24]) and EAU (European Association of Urology) risk assessment[25] (Supplementary Table 2).

Transcriptomic classes were significantly associated with various clinicopathological parameters (Fig. 1e and Supplementary Table 3). Class 2a was enriched for T1 tumors, high-grade tumors, and tumors from patients with CIS and high EORTC risk scores (>6; Supplementary Table 3). Tumors in class 2a also showed a significant overlap with those expressing our previously reported progression-[4,6] and CIS-signatures[5] ($p = 6.4 \times 10^{-28}$ and $p = 1.2 \times 10^{-6}$, respectively; Wilcoxon rank-sum test). Classification using the Lund system[26] revealed that 91% of

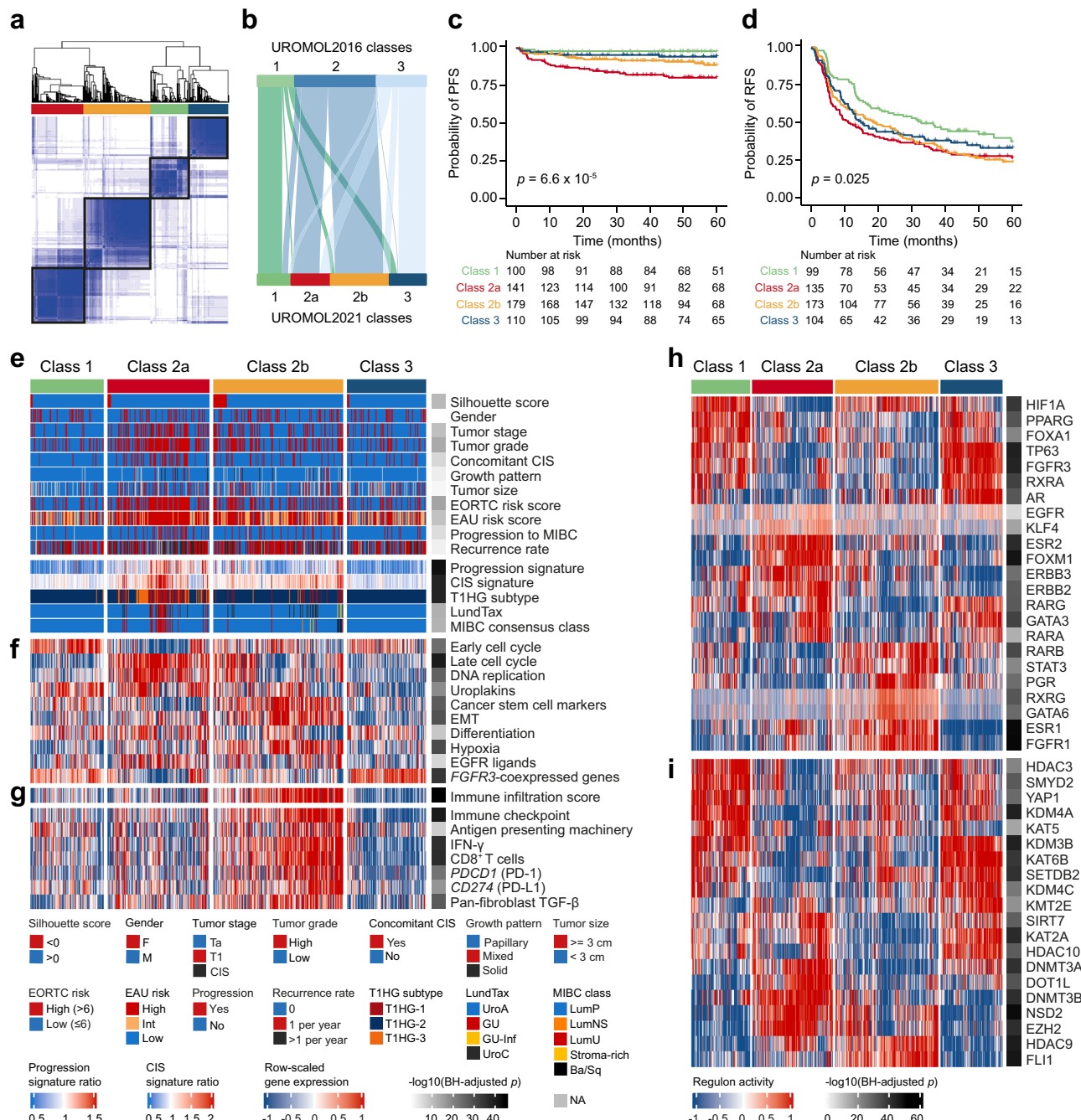

**Fig. 1 Transcriptomic classes in NMIBC. a** Consensus matrix for four clusters. Samples are in both rows and columns and pairwise values range from 0 (samples never cluster together; white) to 1 (samples always cluster together; dark blue). **b** Comparison between the three UROMOL2016 transcriptomic classes and the UROMOL2021 four-cluster solution (76% of tumors in UROMOL2016 class 1 remained class 1, 92% of tumors in UROMOL2016 class 2 remained class 2a/2b and 67% of tumors in UROMOL2016 class 3 remained class 3). **c** Kaplan–Meier plot of progression-free survival (PFS) for 530 patients stratified by transcriptomic class. **d** Kaplan–Meier plot of recurrence-free survival (RFS) for 511 patients stratified by transcriptomic class. **e**, **f** Clinicopathological information and selected gene expression signatures for all patients stratified by transcriptomic class. Samples are ordered after increasing silhouette score within each class (lowest to highest class correlation). CIS carcinoma in situ, EORTC European Organisation for Research and Treatment of Cancer, EAU European Association of Urology, MIBC muscle-invasive bladder cancer, EMT epithelial-mesenchymal transition. **g** RNA-based immune score and immune-related gene expression signatures for all patients stratified by transcriptomic class. **h** Regulon activity profiles for 23 transcription factors. Samples are ordered after increasing silhouette score within each class (lowest to highest class correlation). Regulons (rows) are hierarchically clustered. **i** Regulon activity profiles for potential regulators associated with chromatin remodeling. The most-upregulated regulons within each class are shown. Regulons are hierarchically clustered. P-values were calculated using two-sided Fisher's exact test for categorical variables, Kruskal–Wallis rank-sum test for continuous variables and two-sided log-rank test for comparing survival curves. Source data are provided as a Source data file.

tumors were classified as UroA and 4% as genomically unstable (GU), the latter mostly found in class 2a (Fig. 1e). When classified according to the six consensus classes of MIBC[27], 93% of tumors classified as Luminal Papillary (LumP). Consequently, the classification provided here captures the molecular granularity of NMIBC superiorly to previous strategies.

Analysis of the biological processes associated with NMIBC classes revealed important information discriminating classical histological features from molecular classification and outcome. Confirming our previous findings, class 1 and 3 tumors were associated with early cell cycle genes ($p = 1.1 \times 10^{-15}$ and $p = 0.003$, respectively; Wilcoxon rank-sum test; Fig. 1f). Furthermore, class 3 tumors were characterized by high expression of *FGFR3*-coexpressed genes and a depleted immune contexture (Fig. 1f, g), as previously demonstrated in MIBC and upper tract urothelial carcinoma[28,29]. By contrast, class 2a tumors were mostly associated with late cell cycle genes ($p = 1.3 \times 10^{-33}$; Wilcoxon rank-sum test), DNA replication ($p = 1.1 \times 10^{-20}$; Wilcoxon rank-sum test), uroplakins ($p = 9.1 \times 10^{-7}$; Wilcoxon rank-sum test) and genes involved in cell differentiation ($p = 2.9 \times 10^{-5}$; Wilcoxon rank-sum test), thus indicating that differentiation and proliferation do not show an inverse association. Additionally, class 2b tumors were predominantly associated with expression of cancer stem cell markers ($p = 9.7 \times 10^{-25}$; Wilcoxon rank-sum test) and genes involved in EMT ($p = 7.3 \times 10^{-24}$; Wilcoxon rank-sum test), but a lesser association with cell proliferation (Fig. 1f and Supplementary Fig. 1a).

We estimated the presence of immune cells by deconvolution of RNA-Seq data[30]. Class 2b tumors had a significantly higher total immune infiltration score compared to all other classes ($p = 1.3 \times 10^{-43}$; Wilcoxon rank-sum test), indicating a high level of immune cell infiltration (Fig. 1g). Class 3 tumors had a significantly lower immune infiltration score compared to both class 1 and 2a ($p = 1.8 \times 10^{-7}$; Wilcoxon rank-sum test). Since class 2b tumors showed a favorable PFS compared to class 2a tumors ($p = 0.024$; log-rank test; Supplementary Fig. 1b), we investigated the prognostic impact of immune infiltration irrespective of NMIBC class. The transcriptome-based measure of immune infiltration was, however, not associated with PFS or RFS per se (Supplementary Fig. 1c, d). We also characterized the four classes using gene signatures of potential relevance for different treatment strategies (Fig. 1f, g). Class 2b tumors showed significantly higher expression of immune checkpoint markers and other immune-related signatures compared to all other classes, suggesting that such tumors might be more responsive to immunotherapies[31,32]. However, no difference in BCG failure-free survival was observed between patients with high-grade class 2a or 2b tumors treated with a minimum of six BCG cycles ($n = 54$, Supplementary Fig. 1e).

To explore transcriptomic differences between NMIBC classes further, we analyzed transcriptional regulatory networks (i.e. regulons) for a predefined list of 23 transcription factors previously investigated for MIBC[33] and candidate regulators associated with chromatin remodeling in cancer[34]. This analysis provided confirmation of the biological relevance of a four-subtype classification, as regulon activities were highly associated with transcriptomic classes (Fig. 1h, i). Similar regulon activity patterns were shared by class 1 and 3 tumors, but class 3 tumors differed by having high AR and GATA3 regulon activity. Class 2a tumors were distinctly associated with high FOXM1, ESR2, ERBB2, and ERBB3 regulon activity, while class 2b tumors showed high activity of the ESR1, FGFR1, RARB, STAT3, and PGR regulons. Activity profiles of regulons associated with chromatin remodeling highlighted additional potential regulatory differences between class 1 and 3 tumors, indicating that epigenetic-driven transcriptional networks (e.g., KMT2E,

KAT2A, KAT5, HDAC10 regulons) might be important differentiators of these classes (Fig. 1i and Supplementary Fig. 1f). The potential epigenetic differences between the classes were further supported by an EPIC BeadChip methylation analysis of 29 Ta high-grade tumors, which demonstrated an overall large difference in methylation between samples from different classes (Supplementary Fig. 1g). Furthermore, when comparing class 1 and 3 tumors, it was revealed that gene promoters were less methylated in class 3 (Supplementary Fig. 1h). In total, 12,035 promoter sites were differentially methylated between class 1 and 3 tumors and of these, 97.9% were more methylated in class 1 compared to class 3.

**Transcriptomic subtypes stratified according to pathological features**. To minimize biological confounding arising from pathological and morphological differences, we performed a sub-analysis of Ta low grade tumors ($n = 286$) and T1 high grade tumors ($n = 101$). Analysis of Ta low grade tumors by unsupervised consensus clustering of gene-based expression values restricted to the 2000 genes with highest variation identified four subtypes significantly overlapping with the UROMOL2021 classes ($p = 4.4 \times 10^{-69}$; chi-square test; Supplementary Fig. 2a). The Ta low grade subtypes were, however, not significantly associated with RFS (Supplementary Fig. 2b). A similar analysis of T1 high grade tumors identified three subtypes: one larger group (T1HG-2), enriched for class 1, -2b and -3 tumors, with higher expression of early cell cycle and *FGFR3*-coexpressed genes and relatively good outcome, and two smaller subtypes of class 2a tumors (Supplementary Fig. 3a, b). T1HG-1 and T1HG-3 were both associated with higher expression of genes involved in late cell cycle and DNA replication, but T1HG-1 was enriched for tumors with high progression- and CIS signature scores, the LundTax GU and UroC subtypes and the MIBC consensus class LumU. The T1HG-3 tumors showed higher expression of uroplakins and differentiation markers, indicating a more differentiated subtype (Supplementary Fig. 3a). Five subtypes of T1 tumors (T1BC classes) were recently reported[35] and here we found an overlap between the T1HG subtypes and T1BC classes ($p = 8.7 \times 10^{-5}$; Fisher's exact test; Supplementary Fig. 3a). We built a single-sample T1HG classifier and classified all 535 tumors (Supplementary Fig. 3c). The T1HG subtypes were significantly associated with PFS ($p = 9.8 \times 10^{-12}$; log-rank test; Supplementary Fig. 3d), and multivariable Cox regression analysis showed that T1HG-1 was independently associated with worse PFS and RFS when adjusted for the clinical EORTC- and EAU risk scores (Supplementary Table 2). The T1HG subtypes were not significantly associated with BCG failure-free survival ($n = 55$; $p = 0.54$; log-rank test). The sub-analysis of pathologically homogeneous tumors demonstrates that the UROMOL2021 classes are not mainly driven by differences in histological and morphological features. The T1HG subtypes overlap partially with previously reported biological subtypes and signatures of aggressiveness; however, the increase in biological granularity is not directly translated into better prediction of outcome, since several progression events are missed using the T1HG classifier (Ta progression sensitivity: T1HG-1 + 3 subtype, 24% (7/29); UROMOL2021 class 2a + 2b, 79% (23/29). T1 progression sensitivity: T1HG-1 + 3 subtype, 69% (25/36); UROMOL2021 class 2a + 2b, 89% (32/36)).

**Chromosomal instability is associated with high-risk NMIBC.** To investigate the genomic heterogeneity of NMIBC further, a total of 473 tumor–leukocyte pairs were analyzed using Illumina SNP arrays. Genomic losses/gains and allelic imbalance were derived from raw-segmented total copy number and B allele

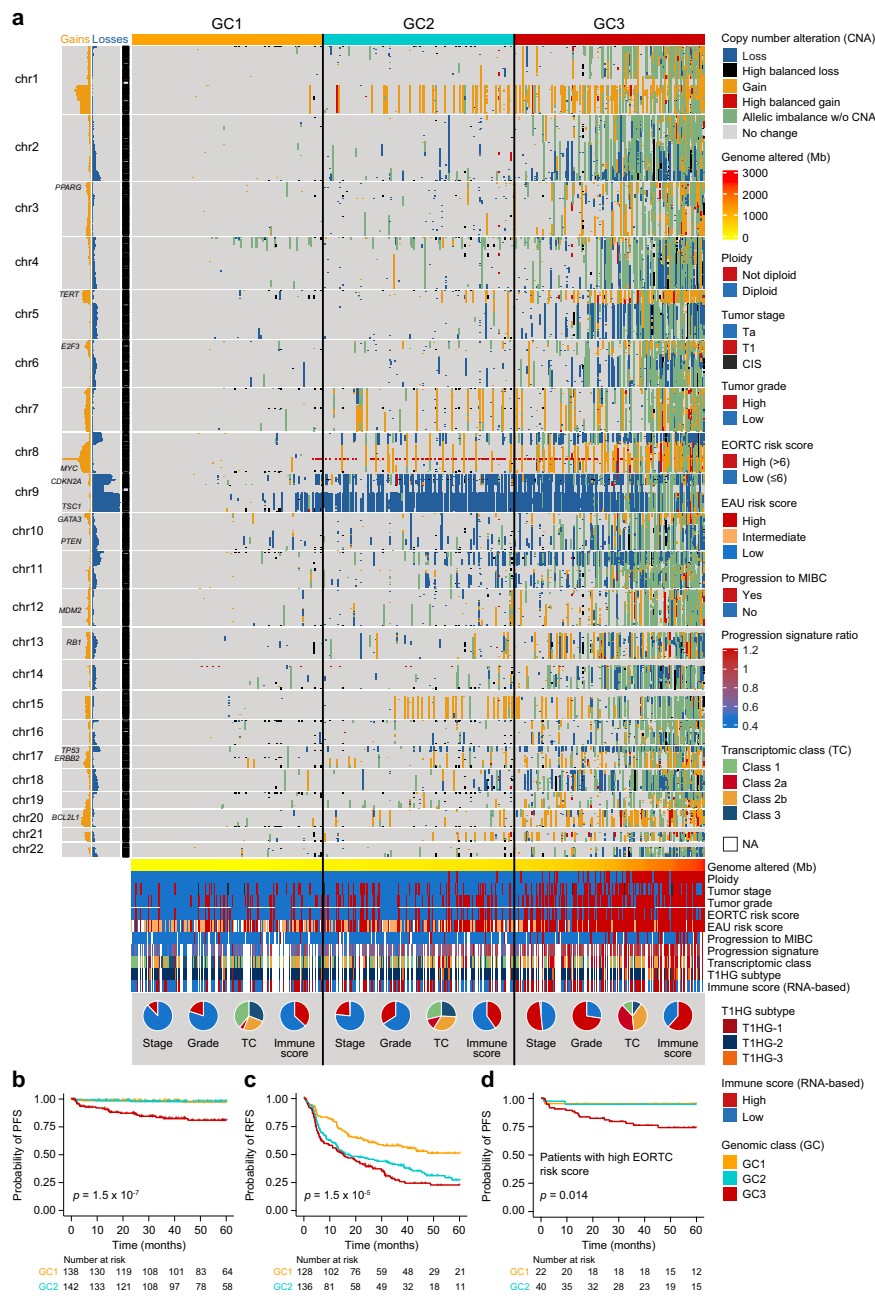

**Fig. 2 Copy number alterations in NMIBC. a** Genome-wide copy number landscape of 473 tumors stratified by genomic class (GC) 1–3. Gains (gain + high balanced gain) and losses (loss + high balanced loss) are summarized to the left of the chromosome band panel. EORTC European Organisation for Research and Treatment of Cancer, EAU European Association of Urology, MIBC muscle-invasive bladder cancer. **b** Kaplan–Meier plot of progression-free survival (PFS) for 426 patients stratified by genomic class. **c** Kaplan–Meier plot of recurrence-free survival (RFS) for 399 patients stratified by genomic class. **d** Kaplan–Meier plot of PFS for patients with high EORTC risk score (n = 163) stratified by genomic class. P-values were calculated using two-sided log-rank test. Source data are provided as a Source data file.

frequency values (for details, see "Methods"). Analysis of the genomic landscape in tumors stratified by EORTC risk score showed similar patterns of abnormalities, but genomic alterations (except for chromosome 9 losses) were more frequently found in EORTC high-risk tumors (Supplementary Fig. 4a). Tumors were therefore stratified to three genomic classes (GC1-3) of equal size with increasing CNA burden to illustrate low, intermediate, and high chromosomal instability (Fig. 2a and Supplementary Fig. 4b). The distribution of clinicopathological parameters and molecular variables between the genomic classes

is shown in Fig. 2a. Specifically, we observed partial or complete loss of chromosome 9 in 53% (251/473; CDKN2A loci) of tumors, and amplification of 8q22.1 in 22% of tumors (103/473; GDF6 and SDC2 loci). Genes in the affected 8q22.1 loci may be involved in the dysregulation of extracellular matrix synthesis and transforming growth factor (TGF)-β pathway[36]. Other frequently altered genomic areas included gains of 1q (16%), 8q (14%; including MYC), 5p (11%; including TERT), 20q (11%) and 20p (9.3%), and losses on 8p (16%), 11p (14%), 17p (13%; including TP53) and 18q (8.2%). Genomic classes were

significantly associated with PFS and RFS ($p = 1.5 \times 10^{-7}$ and $p = 1.5 \times 10^{-5}$, respectively; log-rank test; Fig. 2b, c). Importantly, restricting the survival analysis to tumors with high EORTC risk score ($> 6$), genomic classes were still significantly associated with PFS (Fig. 2d). Genomic classes were significantly associated with stage, grade, concomitant CIS, and EORTC risk score (Fig. 2a and Supplementary Table 4); however, multivariable Cox regression analysis documented that genomic classes were an independent prognostic variable for progression when adjusted for tumor stage and grade (HR = 3.5 (95% CI: 1.57–7.56); $p = 0.002$) and EORTC risk score (HR = 2.8 (95% CI: 1.28–5.99); $p = 0.01$) (Supplementary Table 2). In addition, genomic classes were also independently associated with recurrence when adjusted for EORTC risk score (HR = 1.5 (95% CI: 1.13-2.04); $p = 0.005$).

**Integration of genomic alterations and transcriptomic classes.** Integrative analysis of genomic and transcriptomic data from 303 tumors showed that transcriptomic classes were significantly associated with genomic classes ($p = 2 \times 10^{-11}$; chi-square test; Fig. 3a). Class 2a included the highest fraction of tumors in GC3 (68%; 39/57). To document this association further, we found a strong correlation between genomic classes and 12-gene qPCR progression risk score ($n = 449$, $p = 3.24 \times 10^{-41}$; Kruskal–Wallis rank-sum test), and tumors with a higher progression score were predominantly class 2a and 2b ($p = 1.8 \times 10^{-32}$; Kruskal–Wallis rank-sum test; Fig. 3b). When analyzing class 2a and 2b tumors only, genomic classes were still significantly associated with PFS; all progression events were associated with GC3 tumors ($p = 0.0007$; log-rank test; Fig. 3c). Likewise, when analyzing genomically high-risk (GC3) tumors only, transcriptomic class 2a and 2b were still associated with PFS ($p = 0.036$; log-rank test; Supplementary Fig. 5a). In addition, T1HG subtypes were also significantly associated with genomic classes ($p = 0.006$ and $p = 7.7 \times 10^{-11}$ for T1HG tumors and all tumors, respectively; chi-square tests; Supplementary Fig. 3e, f).

Single-nucleotide variants (SNVs) with moderate or high functional impact were called based on RNA-Seq data. Class 2a tumors showed a significantly higher number of SNVs compared to all other classes ($p = 7.7 \times 10^{-8}$; Kruskal–Wallis rank-sum test; Fig. 3d). Selected frequently mutated genes in bladder cancer are listed in Fig. 3e, and a complete list of the most frequently mutated genes and genes with significantly different mutation patterns across classes can be found in Supplementary Fig. 5b. Mutation calling based on RNA-Seq data has several limitations compared to DNA sequencing, and without a reference germline comparison there is a risk of including germline variants in the analysis. We compared RNA-Seq and whole-exome sequencing (WES) of tumors and germline for 38 patients, and found that the filtering approach applied per sample and across samples enriched significantly for somatic SNVs in our presented gene lists (Fig. 3f). Additional comparative analysis of mutations observed in DNA documented a high correlation between observations in DNA and RNA (Supplementary Fig. 5c–e), suggesting that potentially included germline variants have limited impact on subsequent analyses.

Analysis of hotspot mutations in *FGFR3* (64%), *PIK3CA* (26%), *RAS* (7%), and *hTERT* (79%) based on tumor DNA is shown in Supplementary Fig. 5b. Furthermore, copy number alterations (from SNP microarray analysis) in disease driver genes are highlighted for comparison, and indicate overall loss of *CDKN2A*, significant gain of *PPARG* and *E2F3* in class 2a and loss of *RB1* in class 2a (Fig. 3e). An overview of genomic alterations significantly associated with transcriptomic classes is shown in Fig. 3g. Notably, p53 pathway alterations, observed in 42% of tumors

(127/303; Fig. 3h), were significantly associated with a high CNA burden ($p = 5.9 \times 10^{-20}$; Wilcoxon rank-sum test; Fig. 3i) and class 2a tumors ($p = 2.8 \times 10^{-7}$; Fisher's exact test). Gene expression levels of key molecules in the p53 pathway (*MDM2*, *E2F3*, *TP53*, *ATM*, and *RB1*) were significantly correlated to the observed genomic changes (Supplementary Fig. 5f–j). *TP53* was affected by both copy number change and point mutation in 17 tumors (Fig. 3h), and the majority of these mutations were homozygous (mean variant allele frequency was 0.89 in tumors with copy number change and 0.65 in tumors without). Furthermore, we found a positive correlation between *TP53* variant allele frequency and genomic changes ($R = 0.44$, $p$-value $= 0.027$; Pearson's correlation). Mutations in DNA damage repair (DDR) genes were significantly associated with RNA-derived mutational load ($p = 2.1 \times 10^{-13}$; Wilcoxon rank-sum test; Fig. 3j). This remained significant when *TP53* mutations were excluded from the analysis ($p = 4.4 \times 10^{-11}$; Wilcoxon rank-sum test). In addition, we found a significantly higher mutational load and immune cell infiltration (RNA-Seq based) in GC3 tumors (Fig. 3k, l).

Furthermore, we inferred seven trinucleotide single-base mutational signatures (Supplementary Fig. 5k), and four signatures showed high correlation to signatures previously identified in bladder cancer[33,37,38]: SBS1 (age-related), SBS2 and SBS13 (related to excessive APOBEC activity) and SBS5 (related to *ERCC2* mutations[39]) (Fig. 3e). Class 2a tumors had significantly more mutations in the context of the APOBEC-related signatures (Fig. 3m). Concordantly, high contribution of the APOBEC-related signatures was associated with worse PFS (Supplementary Fig. 5l), indicating that APOBEC activity may drive disease evolution and tumor aggressiveness[8].

Finally, we applied a deconvolution method (WISP; weighted in silico pathology) to assess intra-tumor heterogeneity and class stability from bulk transcriptomic profiles[40]. WISP calculates pure population centroid profiles from the RNA-Seq data and estimates class weights for each sample based on the centroids (hence, each sample is weighted between all four transcriptomic classes; for details, see "Methods"). We found that samples exhibited heterogeneity in all classes, with class 2a having the highest degree of heterogeneity and class 3 the lowest (Supplementary Fig. 6a). Associations of WISP class weights to molecular and clinical features were consistent with the previous description of the classes (Supplementary Fig. 6b–d). Class 1 weights were associated with lower tumor stage, tumor size, and EORTC risk score. Class 2a weights were associated with *TP53* ($p = 4.51 \times 10^{-9}$; Wilcoxon rank-sum test) and *TSC1* ($p = 1.37 \times 10^{-5}$; Wilcoxon rank-sum test) mutations, as well as to higher tumor stage, tumor grade and EORTC risk score. Class 2b weights were significantly correlated to infiltration by all tested immune- and stromal cell populations ($p$-values ranging from $2.2 \times 10^{-12}$ for endothelial cells to $6.5 \times 10^{-126}$ for B lineage cells; Spearman's correlation; Supplementary Fig. 6d). In addition, class 3 weights were associated with *FGFR3* and *PIK3CA* mutations, as well as lower tumor stage and grade. WISP class weights also outlined differences between class 1 and 3 signals: high class 1 weights were associated to *RAS* mutations and infiltration by myeloid dendritic cells, while high class 3 weights were strongly associated to *FGFR3* mutations ($p = 7.34 \times 10^{-15}$; Wilcoxon rank-sum test) and lower immune and stromal population scores than the other classes.

**Spatial proteomics analysis of tumor and immune cell contexture.** To resolve the immune features described above at the spatial level, multiplex immunofluorescence (mIF) and immunohistochemical (IHC) analyses were performed on 167 tumors,

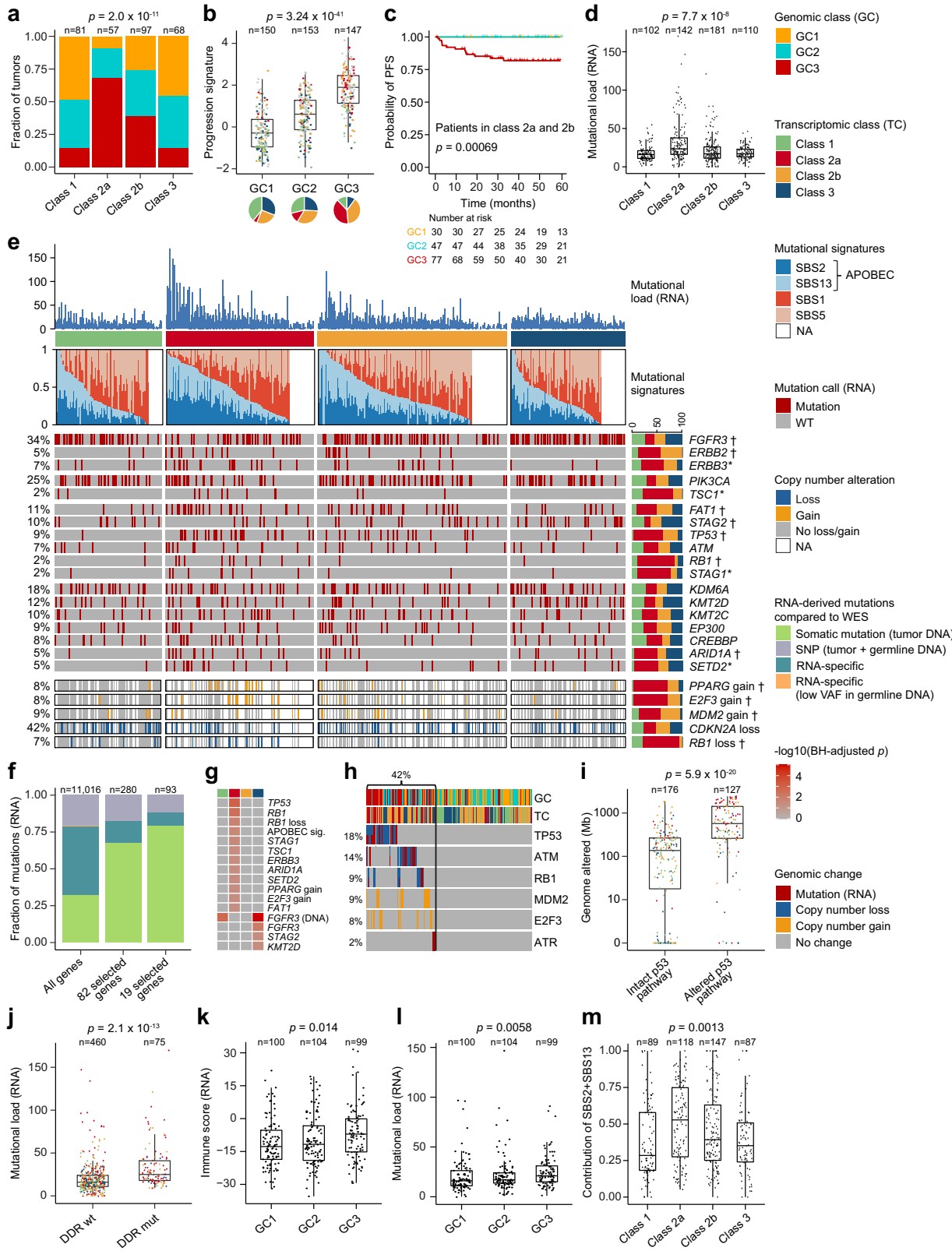

where additional tissue was available. We analyzed immune cells (T helper cells, CTLs, Tregs, B-cells, M1- and M2 macrophages; see Supplementary Fig. 7a for details), carcinoma cells (pancytokeratin, CK5/6, and GATA3) and immune recognition/escape mechanisms (PD-L1 and MHC class I). Automated image

analysis algorithms[41] were applied to study the spatial organization of immune cells and immune evasion mechanisms (Fig. 4a and Supplementary Fig. 7a). RNA-Seq data was available for 150 of the tumors and the RNA-derived immune score correlated significantly with the level of infiltrating immune cells in the

**Fig. 3 Genomic alterations associated with transcriptomic classes. a** Genomic classes (GCs) compared to transcriptomic classes ($n = 303$). **b** 12-gene qPCR-based progression risk score compared to GCs. Colors indicate transcriptomic classes. **c** Kaplan–Meier plot of progression-free survival (PFS) for 154 patients (including only class 2a and 2b tumors) stratified by GC. **d** Number of RNA-derived mutations according to transcriptomic classes. **e** Landscape of genomic alterations according to transcriptomic classes. Samples are ordered after the combined contribution of the APOBEC-related mutational signatures. Panels: RNA-derived mutational load, relative contribution of four RNA-derived mutational signatures (inferred from 441 tumors having more than 100 single nucleotide variations), selected RNA-derived mutated genes, copy number alterations in selected disease driver genes (derived from SNP arrays). Asterisks indicate $p$-values below 0.05. Daggers indicate BH-adjusted $p$-values below 0.05. **f** Comparison of RNA-derived single nucleotide variations to whole-exome sequencing (WES) data from 38 patients for 11,016 mutations in all genes, 280 mutations in the genes most frequently mutated or differentially affected between the classes ($n = 82$, Supplementary Fig. 5b) and 93 mutations in 19 selected bladder cancer genes (Fig. 3e). Only mutations with > 10 reads in tumor and germline DNA were considered and a mutation was called observed when the frequency of the alternate allele was above 2%. **g** Genomic alterations significantly enriched in one transcriptomic class vs. all others. **h** Overview of p53 pathway alterations for all tumors with available copy number data and RNA-Seq data ($n = 303$). **i** Amount of genome altered according to p53 pathway alteration. **j** Number of mutations according to mutations in DNA-damage response (DDR) genes (including *TP53, ATM, BRCA1, ERCC2, ATR, MDC1*). **k** RNA-based immune score according to GCs. **l** RNA-derived mutational load according to GCs. **m** Relative contribution of the APOBEC-related mutational signatures according to transcriptomic class. *P*-values were calculated using two-sided Fisher's exact test for categorical variables, Kruskal–Wallis rank-sum test for continuous variables and two-sided log-rank test for comparing survival curves. For all boxplots, the center line represents the median, box hinges represent first and third quartiles and whiskers represent ± 1.5× interquartile range. Source data are provided as a Source data file.

tumor parenchyma ($p = 1.4 \times 10^{-7}$; Pearson's correlation; Fig. 4b). The different subsets of lymphocytes were predominantly present simultaneously in the stroma and the tumor parenchyma. Consequently, only a few tumors belonged to the immune excluded phenotype[42], and we, therefore, focused on the degree of infiltrating immune cells located in the tumor parenchyma, henceforth termed immune infiltration. Notably, tumors with a high immune infiltration showed a high expression of MHC class I ($p = 6.18 \times 10^{-12}$; Wilcoxon rank-sum test). Only a few tumors expressed PD-L1 in the tumor parenchyma, and the majority of these tumors were highly inflamed ($p = 1.64 \times 10^{-5}$; Wilcoxon signed-rank test; Fig. 4b).

The level of infiltrating immune cells identified at the proteomic level was significantly associated with transcriptomic classes and class 2b tumors showed the highest immune infiltration ($p = 8 \times 10^{-5}$, Kruskal–Wallis rank-sum test; Fig. 4c), supporting the observations delineated from the transcriptomic deconvolution analysis. The differences among transcriptomic classes were particularly evident for T helper cells and CTLs ($p = 2.5 \times 10^{-5}$ and $p = 0.0082$, respectively; Kruskal–Wallis rank-sum test; Supplementary Fig. 7b). These data confirm that the transcriptomic-based estimation of inflammation in class 2b tumors truly represents high immune cell infiltration.

Despite the overall aggressive characteristics of the inflamed class 2b tumors, a high immune infiltration was significantly associated with a lower recurrence rate ($p = 0.022$; Jonckheere–Terpstra test for trend; Fig. 4d), particularly for T helper cells and CTLs ($p = 0.019$ and 0.012, respectively; Jonckheere–Terpstra test for trend Supplementary Fig. 7c). There were too few progression events to document this effect on PFS. Furthermore, a possible protective immune response was shown in patients with tumors of similar genomic background (few genomic alterations); in this group, patients with high immune infiltration had a longer RFS compared to patients with low immune cell infiltration ($p = 0.011$; log-rank test; Fig. 4e).

Finally, we stained for basal cytokeratin expression (CK5/6) and luminal characteristics (GATA3) and aligned these with a pan-cytokeratin staining of the carcinoma cells to estimate the proportion of carcinoma cells positive for CK5/6, GATA3 or both (Supplementary Fig. 7d). All tumors stained positive for GATA3 and 23% for CK5/6 (positivity: > 50% GATA3 or CK5/6 positive cells in the tumor parenchyma). All CK5/6 positive tumors were concurrently GATA3 positive and thereby not basal/squamous by definition[43]. Similar coexpression of basal- and luminal-like markers has been observed previously in the Urothelial-like B tumors[44]. The fraction of CK5/6 positive cells was associated with

transcriptomic classes, with class 3 having the strongest enrichment for CK5/6 expression ($p = 4.5 \times 10^{-8}$; chi-squared test; Fig. 4f).

**Integrative prediction models, classifier construction, and independent validation.** An overview of the univariate Cox regression analyses of selected clinical features and molecular variables is shown in Fig. 5a. In addition, we performed receiver operating characteristic (ROC) analysis for predicting progression within five years using logistic regression models ($n = 301$, Fig. 5b). Combining EORTC risk score with genomic classes increased the predictive accuracy from 0.77 to 0.82, and combining EORTC risk score and transcriptomic classes increased the predictive accuracy to 0.85. Including all three variables in the model slightly increased the predictive accuracy to 0.88 (BH-adjusted $p = 0.033$, Likelihood ratio test; full model vs. EORTC model). Logistic regression models including continuous variables (EORTC, genome altered, and 12-gene progression score), EAU risk scores and T1HG subtypes showed no increased predictive value (Supplementary Fig. 8a–d).

Overall, each transcriptomic class has distinct clinical features, molecular characteristics, and tumor microenvironments, as summarized in Fig. 5c. To facilitate the use of the four transcriptomic classes in future research and clinical settings, we constructed a single-sample classifier for NMIBC. The classifier was built similarly to the recently published tool for the consensus subtypes of MIBC[27], where a class label is assigned to the transcriptomic profile of a tumor based on correlation to the class-specific mean expression profiles (for details, see "Methods"). We applied the classifier to 14 independent cohorts, including three unpublished datasets, with a total of 1228 patients whose tumors were analyzed with a wide range of platforms (Fig. 6a). Notably, RNA-Seq platforms were better suited to call class 3 tumors compared to microarray analyses. Overall, we found highly significant correlations between class and tumor stage, tumor grade and mutations in *FGFR3* and *TP53* (Fig. 6b), and classes showed significantly different PFS ($p = 0.0002$; log-rank test; Fig. 6c) where patients with class 2a tumors had the worst outcome. Notably, multivariable Cox regression analysis revealed that class 2a (HR = 2.9 (95% CI: 1.53–5.27); $p = 0.0009$) and class 2b (HR = 2.1 (95% CI: 1.01–4.36); $p = 0.046$) were independently associated with worse PFS compared to class 1 when adjusted for tumor stage (Supplementary Table 5). For comparison, the T1HG classifier was also applied to the independent samples. The biological features of the T1HG

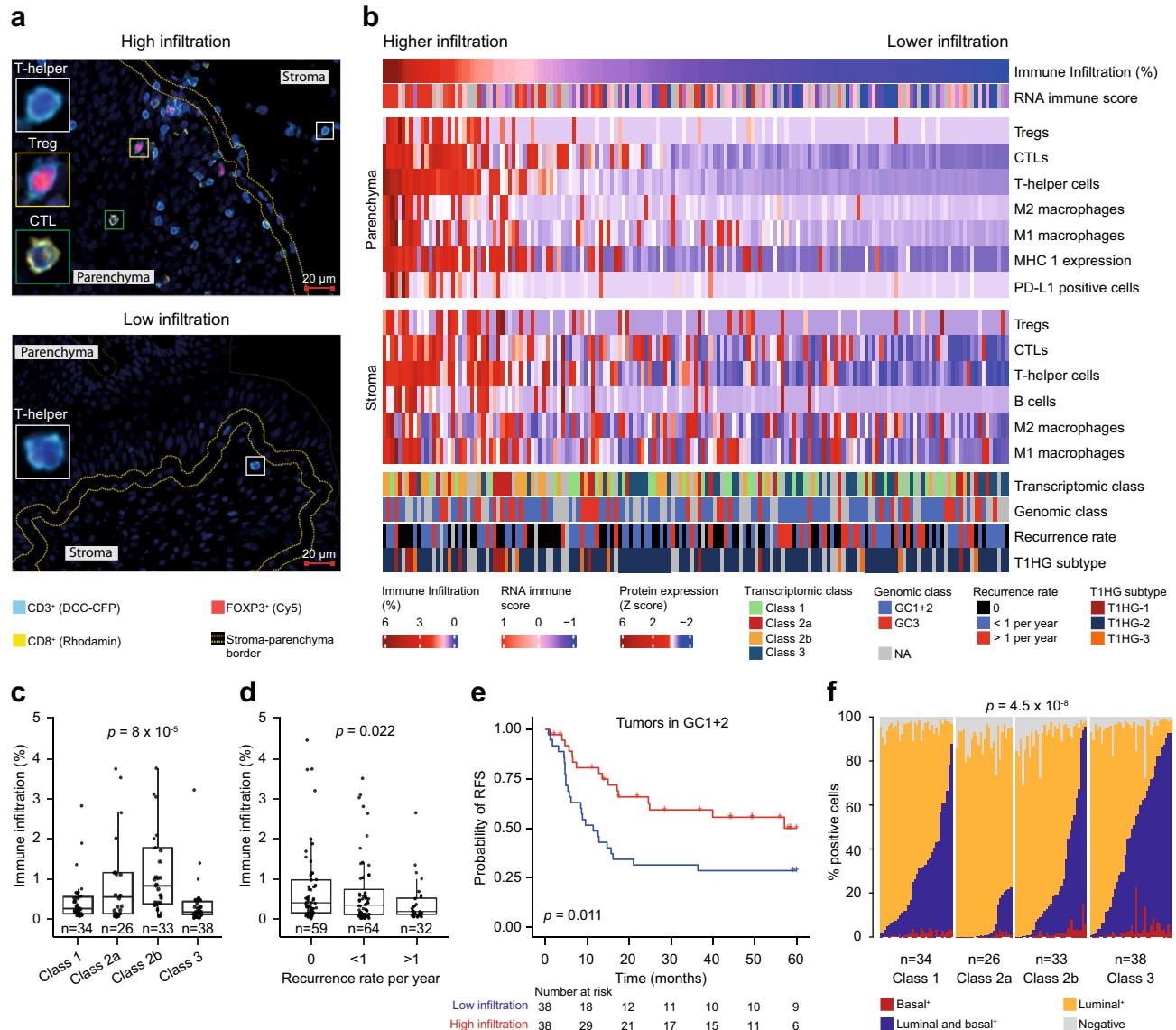

**Fig. 4 Spatial proteomics analysis of tumor immune contexture. a** Multiplex immunofluorescence staining with Panel 1 (CD3, CD8, and FOXP3) of tumors with high- and low immune infiltration with magnifications of T helper cells (CD3+, CD8− and FOXP3−), a cytotoxic T lymphocyte (CTL; CD3+, CD8−, FOXP3−) and a regulatory T cell (Treg; CD3+, CD8− and FOXP3+). Yellow dashed lines divide the tumor tissue into parenchymal and stromal regions. Scale bar: 20 μm. All protein measurements were performed once for each distinct sample. **b** Spatial organization of immune cell infiltration and antigen recognition/escape mechanisms (MHC class 1 and PD-L1) with associated data for genomic class, transcriptomic class, and recurrence rate. The immune cells and immune evasion markers are defined as the percentage of positive cells in the different regions (stroma and parenchyma) and normalized using z-scores, (1) $z = \frac{(x-\mu)}{\sigma}$. Columns are sorted by the degree of immune infiltration into the tumor parenchyma in descending order from left to right. **c** Immune infiltration stratified by transcriptomic class. Immune infiltration is defined as the percentage of total cells in the parenchyma classified as immune cells. The p-value was calculated using two-sided Wilcoxon rank-sum test. **d** Immune infiltration stratified by recurrence rate. The p-value was calculated by the one-sided Jonckheere–Terpstra test for trend. **e** Kaplan–Meier plot of recurrence-free survival (RFS) for patients with tumors with few genomic alterations (GC1 + 2) stratified by immune infiltration. P-value was calculated using two-sided log-rank test. **f** Distribution of CK5/6 and GATA3 positive carcinoma cells stratified by transcriptomic class. Each column represents a patient. The p-value reflects the difference in CK5/6 expression across classes and was calculated by chi-squared test. For boxplots, the center line represents the median, box hinges represent first and third quartiles and whiskers represent ± 1.5× interquartile range. Source data are provided as a Source data file.

subtypes were consistent when considering all tumor stages (n = 1226) and T1 tumors only (n = 663) (Supplementary Fig. 9a). A significant correlation to outcome was observed for the full cohort (p = 0.001; log-rank test; Supplementary Fig. 9b), but the stratified analysis of T1 tumors did not show a significant association to outcome (p = 0.097; log-rank test; Supplementary Fig. 9c). This underlines our observation from the discovery

cohort, that the T1HG classifier separates patients into biological subtypes that may not be important for clinical outcome.

To validate the UROMOL2021 classifier further, we compared differences of regulon activity and biological pathway enrichment between classes in the discovery cohort to findings in the independent cohorts. The regulon and pathway analysis documented a high concordance between datasets (Fig. 6d, e

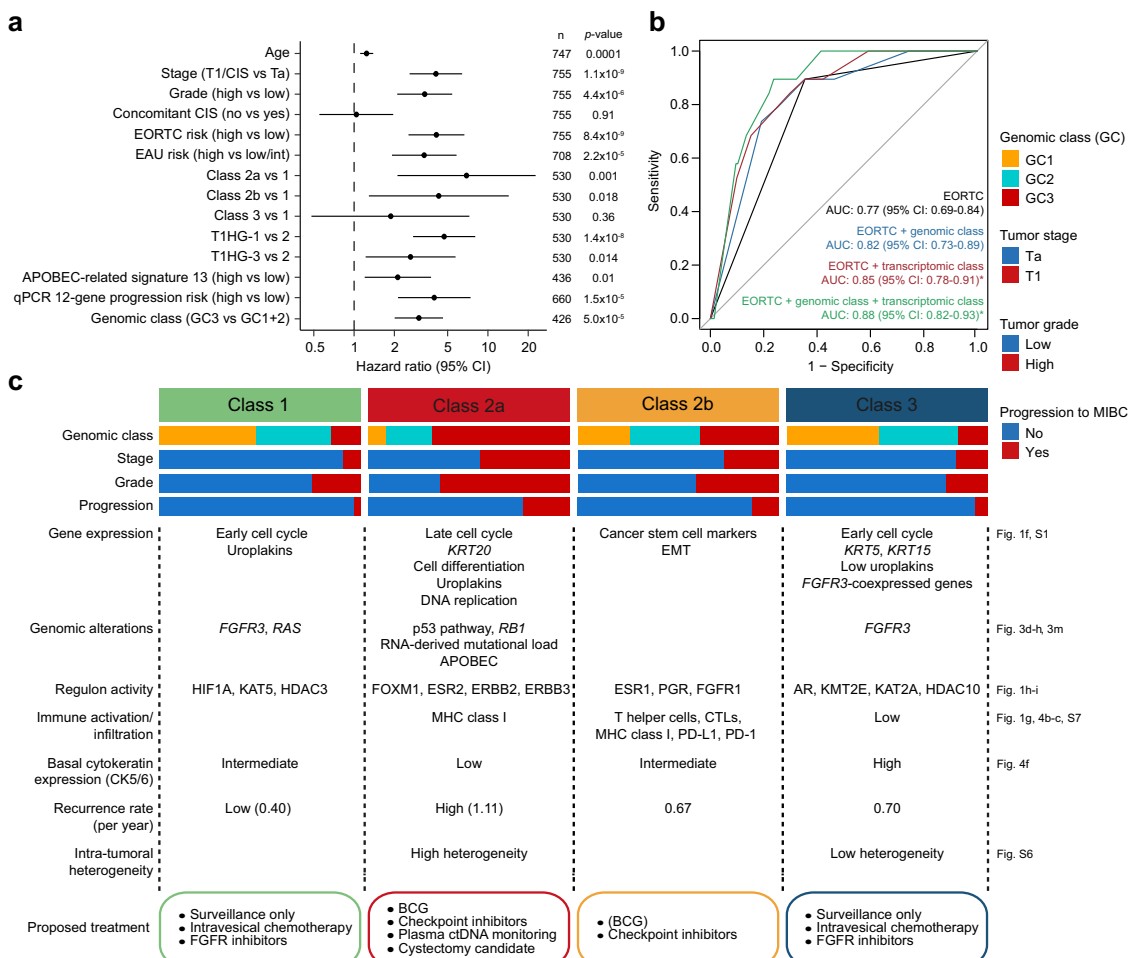

**Fig. 5 Prediction models and summary characteristics of classes. a** Overview of hazard ratios calculated from univariate Cox regressions of progression-free survival using clinical and molecular features. Black dots indicate hazard ratios and horizontal lines show 95% confidence intervals (CI). Asterisks indicate *p*-values below 0.05 and the sample sizes, *n*, used to derive statistics are written to the right. CIS carcinoma in situ, EORTC European Organisation for Research and Treatment of Cancer, EAU European Association of Urology. **b** Receiver operating characteristic (ROC) curves for predicting progression within 5 years using logistic regression models (*n* = 301, events = 19). Asterisks indicate significant model improvement compared to the EORTC model (Likelihood ratio test, BH-adjusted *p*-value below 0.05). AUC area under the curve, CI confidence interval. **c** Summary characteristics of the transcriptomic classes. Molecular features associated with the classes are mentioned, and suggestions for therapeutic options with potential clinical benefit are listed. MIBC muscle-invasive bladder cancer, EMT epithelial-mesenchymal transition, CTLs cytotoxic T lymphocytes. Source data are provided as a Source data file.

and Supplementary Fig. 9d), supporting the robustness of the classes.

## Discussion

Here we expanded our analysis of NMIBC biology and associated clinical outcomes to 834 patient samples from the UROMOL consortium's multicenter study. Utilizing integrative multi-omics analysis, we demonstrated that disease aggressiveness in NMIBC patients was associated with genomic alterations, transcriptomic classes, and immune cell infiltration. We described the development and validation of a single-sample transcriptomic classifier for NMIBC, and identified patients with high chromosomal instability and poor outcome, denoted as class 2a. We demonstrated that the genomic and transcriptomic subtypes showed independent prognostic value when compared to clinical risk factors. Integrative disease models of clinical risk factors and molecular features showed that the addition of transcriptomic

class or genomic instability measures result in similar significant increases in area under the curves (AUCs), and the inclusion of both variables in disease models minimally improved the predictive accuracy (Fig. 5b and Supplementary Fig. 8a). Future classification schemes that incorporate all predictive molecular features may be optimal; however, for clinical application, we suggest the use of a transcriptomic-based classifier, as the expression data in addition will reflect tumor biological processes and possible treatment options (Fig. 5c). The classifier was successfully validated using data from 1228 yet unpublished- and previously published patient samples.

Specifically, we showed that the extent of genomic alterations in NMIBC is an independent predictor of recurrence and progression. Tumors with high chromosomal instability should, therefore, optimally be managed as high-risk tumors regardless of histopathological findings. In addition, we demonstrated that the number of genomic alterations is significantly associated with high-risk transcriptomic classes, p53 pathway alterations, and

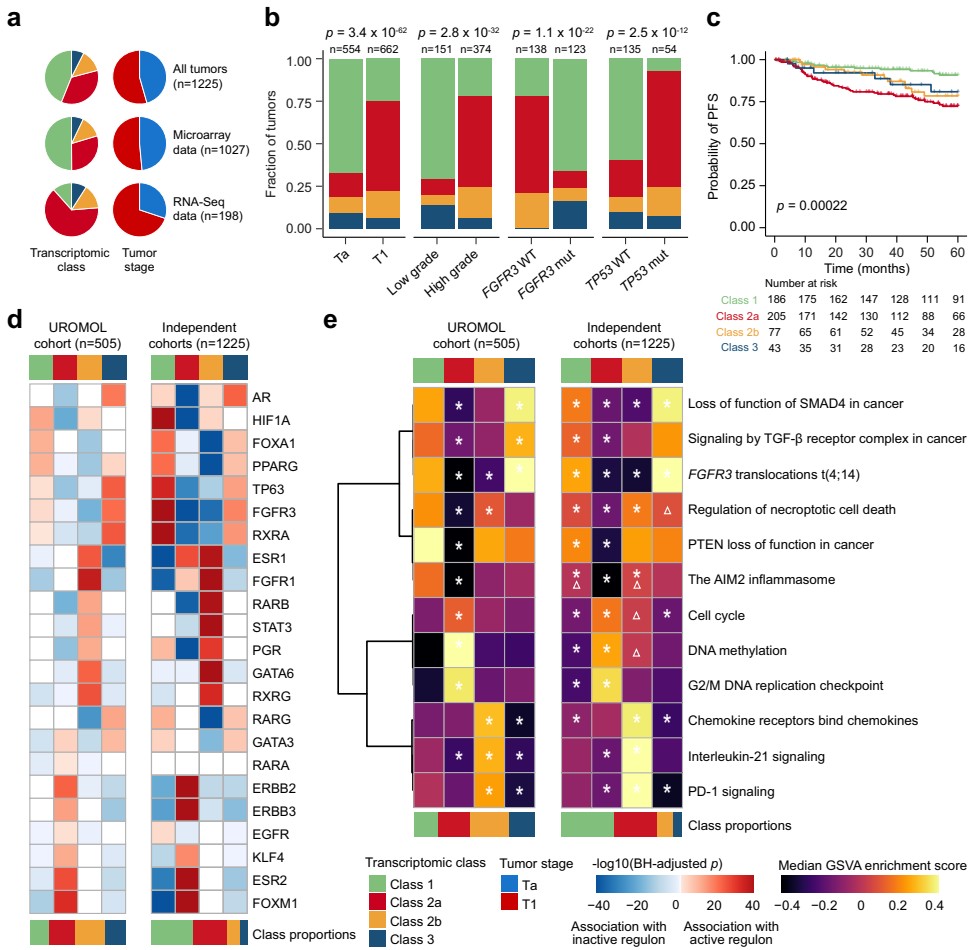

**Fig. 6 Validation of transcriptomic classes in independent cohorts. a** Summary of classification results and stage distribution for all tumors, tumors with microarray data and tumors with RNA-Seq data (1228 tumors were classified in total and 1225 of these were assigned to a class). **b** Association of tumor stage, tumor grade and *FGFR3* and *TP53* mutation status with transcriptomic classes. *P*-values were calculated using two-sided Fisher's exact test. **c** Kaplan–Meier plot of progression-free survival (PFS) for 511 patients stratified by transcriptomic class. The *p*-value was calculated using two-sided log-rank test. **d** Association of regulon activities (active vs. repressed status) with transcriptomic classes in the UROMOL cohort (including samples with positive silhouette scores, *n* = 505) and transcriptomic classes in the independent cohorts (pooled). The heatmap illustrates BH-adjusted *p*-values from two-sided Fisher's exact tests. **e** Pathway enrichment scores within transcriptomic classes in the UROMOL cohort (including samples with positive silhouette scores, *n* = 505) and transcriptomic classes in the independent cohorts (pooled). Asterisks indicate significant association between pathway and class (one class vs. all other classes, two-sided Wilcoxon rank-sum test, BH-adjusted *p*-value below 0.05). Triangles indicate direction swaps of pathway enrichment in the independent cohorts compared to the UROMOL cohort. GSVA gene set variation analysis. Source data are provided as a Source data file.

increased immune cell infiltration. Previous studies that applied array-based CGH analysis have shown that genomic alterations were correlated to histopathological parameters such as stage and grade[13,14,45], but independent prognostic value has not previously been described. We investigated a large clinically well-annotated patient series and applied SNP array technology to increase the granularity of genomic analysis and to gain information of allelic imbalance—a molecular feature not available through CGH analysis. In the current study, we did not aim at identifying specific genomic loci associated with progression to MIBC, but instead we report that the overall CNA burden is directly associated with clinical outcome. This observation is in agreement with other findings linking chromosomal instability to intra-tumor heterogeneity, disease aggressiveness, and poor patient outcome in various human tumor types[46,47]. In bladder cancer, chromosomal instability has previously been linked to advanced

muscle-invasive disease[48]. Our observation is further strength-ened by the identification of mutations in DDR genes and p53 pathway alterations which were associated with genomic instability. This link has been observed previously in a smaller set of bladder tumors[48]. The underlying mechanisms responsible for the genomic instability is, however, not fully understood, but may be caused by oncogene activation and replication stress, which triggers DDR checkpoints[49]. Mutations in DDR genes and p53 pathway alterations are therefore likely to cause the genomic instability observed. We used RNA-Seq data for mutation calling, which is associated with some limitations as only mutations in expressed genes can be detected and no germline reference is used to eliminate germline variations. However, we applied very stringent filters per sample and across the sample cohort to avoid false positives and observed a relatively low number of germline SNPs when comparing to WES in a subset of samples.

Furthermore, a similar approach shown in a recent study also documented that high-precision analysis of mutations based on RNA is achievable[50].

At the transcriptomics level, we identified four main classes of NMIBC with the UROMOL2016 defined class 2 being separated into two groups: class 2a and class 2b. Class 2a, displaying a higher RNA-derived mutational load and elevated APOBEC-related mutational signature contribution, was characterized as a high-risk group with multiple progression events, whereas class 2b, displaying higher expression of stem cell and EMT markers and immune infiltration, was associated with a lower risk of progression. APOBEC-associated mutations are proposed to drive tumor evolution and disease aggressiveness in lung cancer[51,52] and high levels of the APOBEC3B protein have been associated with poor outcome in breast cancer[53]. High tumor mutational burden and APOBEC mutational load have, however, previously been associated with a better prognosis of MIBC[33]. Possible reasons for this discrepancy could be related to better treatment efficacy for muscle-invasive tumors with a high mutational burden and APOBEC-related mutational signatures[3].

The ability to discriminate between class 1 and class 3 tumors was possible since we analyzed a large number of samples by RNA-Seq, a technology with much higher resolution than microarrays used in prior studies. Methylation analysis further emphasized the distinctive features of these two classes (Supplementary Fig. 1g, h). We also observed that class 3 tumors showed a high level of keratin 5 gene expression and simultaneously the highest level of CK5/6 protein expression; however, this should not be associated with basal/squamous MIBC tumors, since we also observed GATA3 expression in all of these tumors (see example in Supplementary Fig. 7d). The transcriptomic classes were prognostic per se, which further highlights several aspects of tumor biology (Fig. 5c). Since all MIBC tumors initially arise as NMIBC, a relevant question is whether the recently developed MIBC consensus classification[27] would be applicable to NMIBC. We provide evidence that this is not the case (Fig. 1e). Our analysis showed that NMIBC displayed less dramatic phenotypic variability compared to MIBC, and classifiers have to be adjusted accordingly. The NMIBC classes described here overlapped partially with previously generated signatures of outcome and gene expression subtypes in NMIBC[6,8,54]. The subtypes from the Lund group initially generated based on the whole spectrum of bladder tumors[7], have now been further developed to include five major tumor cell phenotypes[26,55]. As the classification system spans a large biological range (NMIBC to MIBC), it may not fully capture the subtype granularity observed exclusively in NMIBC. In our work, we compared our transcriptomic classes to the Lund classes using the Lund single-sample classification system[55]. Although we observed an overlap between e.g., class 2a and GU, most tumors were classified as UroA.

Our analysis of regulons revealed potential druggable pathways related to sex hormones in distinct tumor subsets: the androgen receptor pathway was significantly activated in class 3 tumors, although there was no enrichment for male patients in this group. In a recent study, low levels of the androgen receptor was linked to increased translation and tumor proliferation in prostate cancer[56], and the high levels observed in class 3 could therefore have a protective effect. Class 2a was dominated by high levels of ESR2 regulon activity, while class 2b was dominated by high levels of ESR1 and PGR, indicating that hormonal receptor activity may play a pivotal role in disease development. The estrogen and androgen receptors have been linked to urothelial tumorigenesis in animal models[57−59]. Furthermore, the androgen receptor has been shown to be expressed in early-stage

bladder tumors[60], corroborating the finding of a unique class with androgen receptor activity in NMIBC. It is, however, important to emphasize that the transcriptomic analyses of regulon activity were based on bulk tumor analysis and some regulon activities could therefore be driven by different tissue compositions, e.g., higher immune infiltration in class 2b tumors.

The different biological characteristics of the transcriptomic classes suggest that specific therapeutic interventions may have different effects in these patients, as outlined in Fig. 5c. Of note, class 2a tumors were characterized by a high RNA-derived mutational load, which is considered to result in an elevated neoantigen burden, and these patients may therefore benefit from immunotherapy. Checkpoint inhibitors have been shown to be most effective in tumors with high mutational burden[31]. Class 2b tumors were frequently PD-L1 positive, suggesting that these patients may also benefit from checkpoint inhibitor immunotherapy, since high PD-L1 has been linked to an improved response to both PD-1 and PD-L1 inhibitors in MIBC[61,62]. The interest toward the use of systemic immunotherapy in NMIBC has gained momentum, and Pembrolizumab (PD-1 inhibitor) has recently been approved by the FDA for high-risk BCG-unresponsive NMIBC patients. The frequent *FGFR3* mutations observed in class 1 and 3 suggest that FGFR inhibitors could be effective in these tumors, especially since the oral FGFR inhibitor BGJ398 recently showed antitumor activity in a marker lesion study of patients with NMIBC (ref. [63], NCT02657486), and Pemigatinib (FGFR1,2,3 inhibitor) is being tested in an ongoing phase II clinical trial in patients with recurrent low- or intermediate-risk tumors (NCT03914794). Intravesical chemotherapy should be considered especially for class 3 tumors, but possibly also for class 1 tumors although the recurrence rate is lower in these patients.

BCG response mechanisms have been studied intensely[64] and so far, one of the most promising markers of BCG response is fluorescence in situ hybridization (FISH, Urovysion) analysis of chromosomal abnormalities[65]. A recent study of resistance to BCG treatment showed a higher baseline tumor PD-L1 expression among patients unresponsive to BCG compared to patients responsive to BCG treatment[66], indicating that the pre-treatment tumor microenvironment may play a crucial role in BCG response mechanisms. Thus, class 2b tumors, with the highest PD-L1 expression, may respond poorly to BCG. In this study, we did not observe any tumor-centric biological variables that were associated with BCG treatment response. However, the number of patients that received >5 cycles of BCG in connection with the analyzed tumor was low, and larger studies of BCG response are needed to delineate response mechanisms.

In conclusion, we report an integrative multi-omics analysis of NMIBC tumors from a total of 834 patients included in the UROMOL project. We delineate biological processes associated with disease aggressiveness based on detailed, high-quality clinical data, and we provide and validate a classification tool for assigning transcriptomic class and associated progression risk to independent samples. Transcriptomic classification of disease biology provides a framework for biomarker discovery in next-generation clinical trials to optimize the current clinical management of patients with NMIBC.

## Methods

**Patients and data in the UROMOL discovery cohort**. Patients in the discovery cohort were included in the UROMOL project and followed according to national guidelines. Further details regarding samples, procedures, and clinical follow-up are listed in ref. [8]. Informed written consent to take part in research projects was obtained from all patients, and all ethical regulations for work with human participants were followed. The study was approved by the Central Denmark Region

Committees on Biomedical Research Ethics (#1994/2920; Skejby, Aalborg, Frederiksberg); the Danish National Committee on Health Research Ethics (#1906019), the ethics committee of the University Hospital Erlangen (#3755); the ethics committee of the Technical University of Munich (#2792/10); Medical Ethics Committee of Erasmus MC (MEC#168.922/1998/55; Rotterdam); the Uppsala Region Committee on Biomedical Research Ethics (#2008/252); the Ethical Committee of Faculty of Medicine, University of Belgrade (#440/VI-7); the Ethics Committee (CEIC) of Institut Municipal d'Assistència Sanitària/Hospital del Mar (2008/3296/I); the ethics committee of the University Hospital Jena (#4774-4/16).

RNA-Seq data from 438 tumors included in our previous work[8] was reanalyzed together with new RNA-Seq data from 97 tumors.

Based on the discovery samples, we created a "BCG cohort" of 55 patients who meet the following criteria: (1) indication of BCG treatment was high-grade disease, (2) the patient received a minimum of six BCG series and (3) BCG treatment was initiated within 12 months after TURB (hence, BCG was given in relation to the analyzed tumor). The BCG cohort was used to investigate time to BCG failure using multiple features available from our datasets. BCG failure-free survival was defined as time to first high grade tumor or first progression to MIBC after BCG treatment.

**DNA and RNA extraction.** Tumor tissue was collected fresh from resection in each clinical center, embedded in Tissue-Tek O.C.T. and snap frozen in liquid nitrogen before storage at −80 °C. Total RNA was extracted from serial cryosections using RNeasy Mini Kit (Qiagen) and quantified using an Infinite 200 PRO NanoQuant spectrophotometer (Tecan). RNA integrity was assessed using a 2100 Bioanalyzer (Agilent Technologies) and only samples with an RNA Integrity Number (RIN) above five were included. DNA was extracted using Puregene DNA Isolation kit (Fischer Scientific).

**Total RNA-Sequencing.** For the 438 tumors included in our previous work[8], library preparation was performed using ScriptSeq (EpiCentre) followed by sequencing on an Illumina HiSeq 2000. For the 97 tumors added in this study, library preparation was performed with KAPA RNA HyperPrep Kit (RiboErase HMR; Roche) using 500 ng input. Libraries were paired-end sequenced using an Illumina NovaSeq6000.

**Gene expression quantification and normalization.** We remapped and requantified all new and previously generated expression data. Salmon[67] was used to quantify the amount of each transcript using annotation from GRCh38. The R packages tximport and edgeR were used to summarize the expression at gene-level and normalize the data, respectively.

**Consensus clustering.** The expression matrix was filtered to only include transcripts with a median expression above zero. Genes were ranked based on median absolute deviation (MAD) across all samples and divided into subsets of the top -2000, -4000, -6000, -8000, -10,000, -12,000 MAD-ranked genes. Consensus clustering was performed on the different gene subsets using the R package ConsensusClusterPlus (settings: maxK = 10, reps = 1000, pItem = 0.95, pFeature = 1, clusterAlg = "hc", distance = "pearson"). To identify the most representative samples within each cluster, silhouette scores were computed for all samples using the R package CancerSubtypes. A four-cluster solution based on the top-4000 MAD-ranked genes was chosen. Consensus clustering was furthermore performed on Ta low grade tumors only (n = 286) and T1 high grade tumors only (n = 101) to identify subtypes within pathologically homogeneous tumors.

**Gene expression signatures.** We extracted genes associated with cell cycle, uroplakins, cancer stem cells, epithelial-mesenchymal/mesenchymal-epithelial transition, and differentiation[7,68,69] and summarized each biological process as the mean expression of all marker genes associated with the given process. Gene expression signatures of bladder cancer have previously been reported, including a progression- and CIS signature[4,5,70,71]. We calculated a progression signature score for all 535 samples in the RNA-Seq cohort as the ratio between the mean expression of genes upregulated in the signature (*KPNA2, BIRC5, UBE2C, CDC25B, MSN, COL4A1, COL18A1*) and the mean expression of genes downregulated in the signature (*COL4A3BP, NEK1, MBNL2, SKAP2, FABP4*). Likewise, we calculated a CIS signature score for all 535 samples in the RNA-Seq cohort as the ratio between the mean expression of genes upregulated in the signature (*IL13RA1, FBXL5, ARL5A, CXCR4, F13B, SHOC2, IL6ST, HLA-DQA1, SPOP, EFEMP1, DCN, COL15A1, LYZ, SPARC, IGKC, TCF4, KRAS, SDCBP, COL3A1, FBXW2, PDGFC, SGCE, BIRC2, GAPVD1, FLNA, PPP2R5C, LUM, MBD4, UAP1, TOP2A, RARRES1, CLIC4, KPNA2*) and the mean expression of genes downregulated in the signature (*FGFR3, LAMB3, ANXA10, CRTAC1, TMPRSS4, CTSE, MST1R, FABP4, CA12, ITGB4, TNNI2, ST3GAL4, PKP1, BCAM, NDUFA4L2, TRIM29, SH3BP1, LTBP3, LYPD3, CDH11, BST2, EEF1A2, CLCA4, BMP7, AKR1B10, KCTD12, KYNU, UPK2, CFD, TMEM45A*). Finally, we characterized the classes using gene signatures of potential relevance for different treatment strategies[7,27,31,32,72–75]. All gene lists can be found in Supplementary Table 6.

**RNA-based estimation of immune cell infiltration.** As in Rosenthal et al.[30], we evaluated immune cell infiltration based on the expression of predefined gene lists for 14 different immune cell populations[76] (for CD4[+] T cells[77]). Gene lists can be found in Supplementary Table 7. A score for each cell type was calculated as the mean expression of all marker genes associated with the given cell type, and a total immune score was defined as the sum of all immune cell type scores.

**RNA-based mutation calling.** Single base mutations were called from the RNA-seq data using the GATK pipeline. Indels were not considered here due to technical issues that may arise from calling this from RNA-Seq data. Briefly, STAR v2.7 was used to align the raw RNA reads to the hg38 human genome assembly and PICARD tools were used to mark duplicates. GATK tools, SplitNCigarReads, BaseRecalibrator, and ApplyBQSR were applied in order to reformat some of the alignments that span introns and correct the base quality score. Finally, the HaplotypeCaller software was used to call variants. The resulting VCF files were annotated using SnpEff followed by filtration for possible impact on proteins. First, only SNVs annotated with a HIGH or MODERATE impact by SnpEff were included and SNVs in splice-site genomic locations were excluded. Second, mutations with an rs ID in dbSNP were excluded. Third, only mutations with a quality score above 100 and a Fisher Strand score (FS) below 30.0 were included. Finally, mutations called in ten or more samples were filtered out with the exception of known mutation hotspots (*FGFR3* and *PIK3CA*). When calculating the RNA-derived mutational load, we excluded mutations that were significantly found more often in samples sequenced on an Illumina NovaSeq 6000 (new additional RNA-Seq data) compared to samples sequenced on an Illumina HiSeq 2000 (original RNA-Seq data) (Fisher's exact test p-values < 0.01), as the samples sequenced using the NovaSeq platform contained considerable more reads. Thereby, a total of 791 genes were considered for the RNA-derived mutational load. Furthermore, we validated RNA-derived mutations in DNA for a subset of patients (n = 38) where whole-exome sequencing data was available. Mutations with > 10 reads in tumor and germline DNA were considered and a mutation was called observed when the frequency of the alternate allele was above 2%.

**RNA-based mutational signature analysis.** To infer mutational signatures, we included mutations called within the gene sequence (HIGH, MODERATE, and LOW impact) and excluded mutations with rs ID together with mutations with a quality score below 100 or a Fisher Strand score (FS) above 30.0. Finally, mutations were included if they met the following criteria: (1) alternate allele frequency (AF) > 0.15 and < 0.60; (2) number of reads > 20. Only samples with more than 100 SNVs were kept to infer the mutational signatures (n = 441). We used non-negative matrix factorization to decompose the motifs matrix into seven signatures and their corresponding weights using the R package SomaticSignatures. The similarity between the seven inferred signatures and defined COSMIC signatures was examined using the R package MutationalPatterns.

**Copy number analysis.** GSA Illumina SNP arrays (~760 k positions) were used on tumor DNA from 473 patients in order to assess copy number alterations. We previously applied the Infinium OncoArray-500K BeadChipGenotyping arrays for the paired germline samples and used this as reference. LogR Ratio (LRR) and B-allele-fraction (BAF) were corrected and normalized using the Genotyping module from GenomeStudio 2.0 (Illumina) within each array type and all positions uniquely found in both arrays were exported for further analysis (151,291 probes). The R package ASCAT was used for segmentation of the genome and we used the raw-segmented total copy number, the raw-segmented BAF data and various empiric thresholds (gains: > 0.08, high gains: > 0.16, loss: < −0.1, high loss: < −0.2, allelic imbalance (AI): < 0.45) to identify five different types of CNAs: (1) losses associated with AI (i.e., associated with a deviation in BAF), (2) gains associated with AI, (3) high losses without AI, (4) high gains without AI and (5) AI without a change in total copy number. The applied thresholds were validated with histograms of LRR and in all diploid cases (83%), the peak for no change in copy number was within the thresholds defined for the gains/losses without deviation in BAF. Using these thresholds, subclonal events present in a minority of carcinoma cells will either not be called or instead be defined as regions with no copy number changes but with deviation in BAF (due to the higher sensitivity of the BAF measurement). Therefore, defined gains/losses are clonal events or subclonal events present in the majority of carcinoma cells.

The amount of genome in a non-normal state was calculated using the thresholds above (referred to as the CNA burden). Tumors were assigned to three genomic classes (GC1-3) of equal size based on the CNA burden to illustrate low, intermediate, and high chromosomal instability (cut-offs at the 33rd and 67th percentiles). Furthermore, based on LRR and BAF plots, we manually defined tumors as being diploid or not diploid.

**Methylation analysis.** DNA methylation analysis was performed using DNA from 29 patients based on the UROMOL2016 classification with 10–11 samples from each class. After re-classification, we had 10 samples in class 1, 12 in class 2a/2b with a majority in class 2b and 6 in class 3. All tumors were selected to have a high silhouette score, and all were Ta high-grade tumors. We used 500 ng genomic DNA

for bisulfite conversion followed by whole-genome amplification prior to hybridization to EPIC BeadChip (Illumina, San Diego, CA) overnight as described by the manufacturer and then scanned with the Illumina iSCAN system. Data was imported and processed using the RnBeads v2.2R package pipeline. For the pre-processing of the data, the normalization method was set to "illumina" and the background correction method to "methylumi.noob".

**Regulon analysis**. We reconstructed transcriptional regulatory networks (regulons) using the R package RTN[78]. We investigated 23 transcription factors previously associated with bladder cancer[33] and 78 candidate regulators associated with chromatin remodeling in cancer[34]. Gene lists can be found in Supplementary Table 8. Potential associations between a regulator and all possible target genes were inferred from the expression matrix by Mutual Information and Spearman's correlation, and permutation analysis was used to remove associations with a BH-adjusted $p$-value $> 1 \times 10^{-5}$. Unstable associations were eliminated by bootstrap analysis (1000 resamplings, consensus bootstrap > 95%) and the weakest association in triangles of two regulators and common target genes were removed by data processing inequality (DPI) filtering (tolerance = 0.01). Regulon activity scores for all samples were calculated by two-tailed gene set enrichment analysis.

**12-gene progression score**. All molecular data related to the qPCR 12-gene progression score were generated previously[6] and analyzed here with additional follow-up information. In brief, RNA was extracted from serial cryosections using an RNeasy Mini Kit (Qiagen). A total of 500 ng RNA was used for cDNA synthesis and PCR amplification was performed using a 7900HT PCR system (Thermo Fisher Scientific). Scores for progression were calculated using non-normalized cycle threshold (Ct) values: mean Ct (*COL4A3BP, NEK1, MBNL2, SKAP2, FABP4*) —mean Ct (*KPNA2, BIRC5, UBE2C, CDC25B, MSN, COL4A1, COL18A1*). Primer sequences for the 12 genes can be found in Supplementary Table 9.

**Construction of single-sample transcriptomic classifier**. We constructed a Pearson nearest-centroid classifier for NMIBC based on the recently published classifier for the MIBC consensus subtypes[27]. Only samples with positive silhouette scores were used for feature selection ($n = 505$). We filtered the expression matrix to include genes with a median expression > 0 in at least one of the four classes and used a step-wise ANOVA approach to identify genes with significantly different expression levels across classes. ANOVA between all four classes resulted in 13,650 significant genes (BH-adjusted $p$-values < 0.05). Genes highly expressed in class 2b dominated the list, so we removed class 2b samples and previously significant genes from the dataset and performed a second round of ANOVA on the remaining classes. This analysis added only four significant genes to the feature list (BH-adjusted $p$-values < 0.05). Next, class 2a samples were removed and one last round of ANOVA between class 1 and class 3 was performed (corresponding to a $t$-test), resulting in 109 significant genes (BH-adjusted $p$-values < 0.05). Thereby, a total number of 13,762 genes were suggested to be differentially expressed between classes. The step-wise ANOVA approach was chosen instead of multiple pairwise $t$-tests to reduce the number of statistical tests while still accessing differences between all classes. We computed the AUC associated with each gene for prediction of the four classes and kept genes with an AUC > 0.6 ($n = 10,149$). An additional filtering of genes was performed to only keep genes with a mean expression > 0 across all samples. Overall, the initial selection of features resulted in a list of 9,451 genes.

We used leave one out cross-validation (LOOCV) to assess the classification performance associated with different subsets of the 9451 features. In each LOOCV run, we computed the mean fold-change associated with each gene for each class versus the others. Genes were ordered by their mean fold-change within each class and the four gene lists were used to generate several gene subsets. The N top upregulated and N top downregulated genes within each class, with N varying from 50 to 800, were selected and used as feature input for the classifier. We obtained the lowest LOOCV error rate when selecting the 368 top upregulated and 368 top downregulated genes within each class (1964 unique genes in total). Finally, genes appearing in > 80% of the LOOCV runs were selected and used to build the final classifier ($n = 1942$). We computed four centroids corresponding to the four NMIBC classes (i.e., the mean gene expression profile of the 1942 chosen feature genes for each class), and class labels are then assigned to single NMIBC samples based on the Pearson correlation between a sample's expression profile and the four-class centroids. The NMIBC classifier is available as a web application at http://nmibc-class.dk, as an R package at https://github.com/sialindskrog/classifyNMIBC or in Supplementary Software 1.

A similar approach was used to construct a Pearson nearest-centroid classifier for the T1HG subtypes, resulting in 883 chosen feature genes.

**Proteomics**. Formalin-fixed paraffin-embedded (FFPE) tissue from transurethral resection of bladder tumors (TURB) was obtained from 167 Danish patients at Skejby and Frederiksberg hospital. Tissue microarrays (TMAs) were constructed from representative tumor areas with 1 mm triplicate core biopsies using the automated TMA-GRAND Master (3DHISTECH Ltd, Budapest, Hungary).

**Immunofluorescence, immunohistochemistry, and imaging**. Multiplex immunofluorescence analysis (mIF) was performed on two TMA sections (3 μm) for detection of Panel 1 (CD3, CD8, and FOXP3) and Panel 2 (CD20, CD68, CD163, and HLA-A, B, C) as in Taber et al.[41]. Stainings were performed on the Discovery ULTRA staining instrument (Ventana Medical Systems), all primary antibodies are listed in Supplementary Data 1. We deparaffinized TMA sections using the EZ Prep solution (Ventana Medical Systems, cat # 950-102) for 16 min at 72 °C. Afterwards, heat-induced epitope retrieval using CC1 solution (Ventana Medical Systems, cat# 950-124) was run for 64 min at 95-100 °C. Then, endogenous peroxidase activity was blocked using a DISC inhibitor reagent (Ventana Medical Systems, cat#760-4840). For fluorescent detection, we utilized a tyramide signal amplification strategy with horseradish peroxidase (HRP)[79]. The first primary antibody (1-ab) was incubated followed by detection using a secondary antibody (2-ab) conjugated with HRP (listed in Supplementary Data 1). Two rinses with reaction buffer (Ventana Medical Systems, cat# 950-300) was then carried out followed by adding and incubating the tyramide conjugated fluorophore (TcF, listed in Supplementary Data 1) for 4 min. We then applied 0.01 % H2O2 (DISCOVERY reagent, Ventana Medical Systems, cat#760-244), and let the TcF react with the HRP in the 1-ab/2-ab complex for 8 min. Heat-mediated stripping of the antibodies was run for 20 min at 100 °C using a CC2 buffer (Ventana Medical Systems, cat#950-223). The cycle was then repeated sequentially with a new 1-ab/2-ab complex and TcF in the order listed in Supplementary Data 1. Afterwards, the TMA sections were counterstained with VECTASHIELD anti-fade mounting medium with DAPI (Ventana Medical Systems, cat#H-1200) for nuclear detection. The fluorophore-labeled sections were imaged at 20× magnification using the NanoZoomer s60 scanner (Hamamatsu Photonics KK, Japan). Immunostaining for pan-cytokeratin (Clone A1/A3, 1:100, 16 min, Dako Agilent, cat#GA005361-2) as a second layer was performed on all mIF stained sections to outline carcinoma cells. For bright-field detection, we used the Ventanas Detection Kits: ultraView Universal 3,3'-Diaminobenzidine (Ventana Medical Systems, cat#760-500) according to the manufacturer's instructions. We counterstained all TMA sections with hematoxylin II (Ventana Medical Systems, cat#790-2208) for 8 min, followed by Bluing reagent (Ventana Medical Systems, cat#760-2037) for 4 min. The Hamamatsu Nanozoomer 2.0 HT (Hamamatsu Photonics KK, Japan) was used for bright-field imaging.

Identification of PD-L1 expression in the tumor parenchyma was performed using two sequential TMA sections, the first section stained against pan-cytokeratin (Clone A1/A3, 1:100, 16 min, Dako Agilent, cat#GA005361-2) and the second against PD-L1 (Clone Sp263, ready to use, 60 min, Ventana Medical Systems, cat#790-4905). Identification of basal and luminal markers on the carcinoma cells was performed using three sequential TMA sections, stained for pan-cytokeratin (Clone A1/A3, 1:100, 16 min, Dako Agilent, cat#GA005361-2), GATA3 (Clone L50-823, ready to use, 24 min, Ventana Medical Systems, cat#7107749001) and CK5/6 (Clone D5/16 B4, ready to use, 24 min, Agilent/Dako cat#M7237). For bright-field detection, the above-mentioned method was used.

**Digital pathology**. For digital pathology, we utilized the Visiopharm image analysis software version 2018.9.5.5952 (Visiopharm A/S, Hørsholm, Denmark). The Tissue Array module was used to identify and extract individual cores on the TMAs and the Tissue Align module to align the pan-cytokeratin stained image with its corresponding fluorescence image (Supplementary Fig. 7a, step 1). Image analysis protocol packages (APPs) developed by our group[41] were used to automatically: (1) Define parenchymal and stromal regions as pan-cytokeratin positive and negative, respectively (Supplementary Fig. 7a, step 2). Calculate the proportion of immune cell subsets based on co-localization of selected markers (Supplementary Fig. 7a, step 3). Calculate the proportion of PD-L positive cells. In addition, we designed an APP using the Tissue Author module, in order to calculate the proportion of GATA3, CK5/6 or double-positive positive cells. For GATA3 and CK5/CK6 detection, tumors were classified as positive if more than 50% of the carcinoma cells expressed the marker. For all markers, we selected a threshold visually to differentiate between positive and unspecific staining. The threshold was verified by an experienced pathologist. We applied the following scoring algorithm to calculate cell fractions (here shown for T helper cells):

$$\text{Fraction of T helper cells} = \frac{\text{Number of CD3}^+, \text{CD8}^-\text{FOXP3}^- \text{ cells in the parenchyma or stroma}}{\text{Total cell count in the parenchyma or stroma}}$$

(2)

The proportion of infiltrating immune cells was consistent across the 3 tissue cores from the same tumor; average Pearson correlation coefficient: 0.67 (panel 1) and 0.77 (panel 2).

**Independent transcriptomics datasets used for validation**. Transcriptomics data from 11 historical cohorts (Kim[80], Lindgren[45], Sjödahl2012[7], CIT[72], Choi[81], Sjödahl2017[26], Song[82], Sjödahl2019[83], Aarhus microarrays[5,54,84,85], Meeks[35]) were used for the validation of the four-class NMIBC classification and T1HG subtypes. The data were downloaded from GEO or ArrayExpress and annotated with HUGO Gene Symbols. In addition to using the publicly available data, we included data from three yet unpublished cohorts (listed below). Each sample in each cohort ($n = 1228$) was classified using the single-sample classifiers trained using the

UROMOL cohort (1225 and 1226 tumors were assigned a class using the NMIBC classifier and T1HG classifier, respectively).

*Unpublished cohort 1 provided by Margaret Knowles.* Data from Affymetrix Human Transcriptome 2.0 microarrays for 104 stage T1 and 113 stage Ta tumors from the Leeds Multidisciplinary Research Tissue Bank following approval by the research ethics committee 10/H1306/7). All ethical regulations for work with human participants were followed. Total RNA was isolated from frozen tissue sections using a RNeasy Plus Micro Kit and amplified using the Affymetrix GeneChip WT PLUS Reagent Kit. The resulting cDNA was hybridized onto Affymetrix Human Transcriptome 2.0 microarrays. Quality control checks, gene level normalization (using SST-RMA) and signal summarization was conducted using Affymetrix Expression Console Software.

*Unpublished cohort 2 provided by Richard Bryan.* RNA-Seq based analysis of 78 tumors from the West Midlands Bladder Cancer Prognosis Programme (BCPP, ethics approval 06/MRE04/65). All ethical regulations for work with human participants were followed. RNA libraries were prepared using the Truseq Stranded Total RNA with Ribo-zero Gold kit (Illumina) and $2 \times 100$ bp PE sequenced (Hiseq, $n = 26$) or $2 \times 75$ bp PE sequenced (Nextseq, $n = 52$). The data were aligned to GRCh37 and reads counted with STAR aligner (v2.5.2b). Log2(Read count+1) for each gene has been used as input for the class prediction.

*Unpublished cohort 3 provided by Trine Strandgaard.* RNA-Seq based analysis of 47 fresh frozen tumors from patients enrolled at Aarhus University Hospital with high-risk NMIBC, and analyzed following approval by the Danish National Committee on Health Research Ethics (#1708266). All ethical regulations for work with human participants were followed. RNA-Seq data was generated using analysis pipelines described above for the additional samples included in the discovery cohort in this work.

**Pathway enrichment analysis**. Pathway enrichment analysis was performed independently in the UROMOL cohort and each historical cohort that contained representatives of all classes. First, we collected pathway annotation from the Reactome (using R package reactome.db v1.68.0) and KEGG (using the R package KEGGREST v1.24.1) databases. We joined these annotations and performed gene-set variation analyses (using the R package GSVA v1.32.0) to obtain single-sample enrichment scores for each pathway.

To find associations between pathways and classes, we performed Mann–Whitney $U$-tests using the pathway enrichment scores between samples in each class versus samples in other classes in each cohort separately. $P$-values were BH-adjusted. For the pathway visualizations, we first filtered pathways that were enriched in the same class in the UROMOL cohort and in at least four other datasets and then manually selected pathways from the filtered list. Pathway enrichment scores were grouped using hierarchical clustering with correlation distances $(1 - r)$ and Ward clustering using the enrichment scores in the UROMOL cohort and the same pathway order was then used for the independent cohorts.

**Regulon activity in validation cohorts**. The regulons from the transcriptional networks calculated from UROMOL data were used to derive differential enrichment scores in each cohort separately using the two-tail GSEA method (R package RTNsurvival). We discretized the activity scores into "active" and "repressed" status, aggregated the regulon status in all cohorts, and used Fisher's exact tests to find the association of regulon status with each class. $P$-values were BH-adjusted.

**Weighted in silico pathology (WISP) analysis**. To approximate intra-tumor heterogeneity, we used the bulk transcriptomic profiles and the consensus clustering results for the UROMOL cohort and applied the Weighted in silico pathology (WISP, R package v. 2.3) method with default settings. Only samples with a positive silhouette score were used for the WISP analysis ($n = 505$). WISP consists of two main steps: (1) Calculation of pure population centroid profiles and (2) Estimation of pure population weights for each sample. First, WISP selects features for each class by iteratively considering ANOVA $p$-values (FDR adjusted $p$-values < 0.05), AUC scores (AUC > 0.8) and expression log-fold changes between classes, fitting a non-negative least squares model and removing samples considered mixed. A model is then built from the core of pure samples for each class (154 samples were kept as "pure" and 199 top marker genes were included in the centroid profiles). Next, WISP class weights were estimated for all the samples in the cohort ($n = 505$) using the centroid profiles (hence, each sample is weighted between all four transcriptomic classes). We recovered the estimated WISP class weights and used Pearson and Spearman correlations to investigate their association to silhouette scores and MCPcounter immune scores[86], respectively. Finally, we used Wilcoxon rank-sum tests to associate WISP class weights to genetic mutations and clinical variables.

**Quantification and statistical analysis**. Statistical comparisons between groups were performed using the two-sided Wilcoxon rank-sum test or Kruskal–Wallis rank-sum test for continuous variables and two-sided Fisher's exact test or chi-square test for categorical variables. It is stated in the figure legends if tests other than the above-mentioned were applied. Survival analyses were performed using the Kaplan–Meier method and two-sided log-rank tests were used to compare survival curves (R packages survival and survminer). Cox Proportional-Hazards analyses were performed using the R packages survival and survminer. We built logistic regression models to predict progression and used the predicted probabilities as variables in ROC analyses (R packages glmnet and pROC). AUCs and associated 95% CIs (computed with 2000 stratified bootstrap replicates) were calculated using the R package pROC. Likelihood ratio tests were used to assess model improvement (all models were compared to the EORTC model). $P$-values below 0.05 were considered significant across all tests and BH-adjustment of $p$-values was performed to control for multiple testing when necessary (otherwise unadjusted $p$ values are reported). The R packages tidyverse, ggplot2, reshape2, and ComplexHeatmap were used for data analysis and figure creation. All statistical and bioinformatics analyses were performed with R (v3.6.0 or 3.6.1).

**Reporting summary**. Further information on research design is available in the Nature Research Reporting Summary linked to this article.

## Data availability

Raw sequencing and SNP data are deposited and available under controlled access at The European Genome-phenome Archive (EGA), which is hosted by the European Bioinformatics Institute (EBI) and the Centre for Genomic Regulation (CRG). The RNA-Seq data are available under accession code: EGAS00001004693 and the SNP data are available under accession code: EGAS00001004862. The data are available under controlled access at EGA. Due to privacy laws, data will be available following new approvals by ethical committees and data protection agencies. The data release process can be initiated by contacting the corresponding author (lars@clin.au.dk). Processed normalized mRNA read counts are available in Supplementary Data 2, processed proteomics data are available in Supplementary Data 3 and processed EPIC BeadChip methylation data are available in Supplementary Data 4. Data are available within the Article file, Supplementary Information or from the authors upon request. The expression data used for validation are available under the following accession codes: Kim[80], microarray, GEO: GSE13507; Lindgren[45], microarray, GEO: GSE32549; Sjödahl2012[7], microarray, GEO: GSE32894; CIT[72], microarray, ArrayExpress: E-MTAB-1803; Choi[81], microarray, GEO: GSE48075; Sjödahl2017[26], microarray, GEO: GSE83586; Song[82], microarray, GEO: GSE120736; Sjödahl2019[83], microarray, GEO: GSE128959; Aarhus microarrays[5,54,84,85], GEO: GSE3167 and GSE5479; Meeks[35], RNA-Seq, GEO: GSE154261; unpublished cohort 1 provided by Margaret Knowles, microarray, GEO: GSE163209; unpublished cohort 2 provided by Richard Bryan, RNA-Seq, EGA: EGAS00001004358; unpublished cohort 3 provided by Trine Strandgaard, RNA-Seq, EGA: EGAS00001005050. Source data are provided with this paper.

## Code availability

We implemented the transcriptomic classification tool as an R package that is available at https://github.com/sialindskrog/classifyNMIBC.

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

## Acknowledgements
We thank K.K. Cheng, Maurice P. Zeegers, Nicholas D. James, Naheema S. Gordon, Ben Abbotts and Roland Arnold for work involved in generating the Birmingham RNA-Seq validation cohort. S.V.L., F.P., and L.D. are supported by the following funding sources: Aarhus University, The Danish Cancer Biobank, The Health Research Foundation of Central Denmark Region, The Danish Cancer Society. N.M. and F.X.R. are supported by Fondo de Investigaciones Sanitarias (FIS), Instituto de Salud Carlos III, Spain (#PI18/01347), Asociación Española Contra el Cáncer (AECC, # GB28012014). D.J.D. is supported by the following funding sources: RSG 17-233–01-TBE from the American Cancer Society, the W.W. Smith Charitable Trust, the Pennsylvania Department of Health via Tobacco CURE Funds, the Ken and Bonnie Shockey Fund for Urologic Research and the Bladder Cancer Support Group at Penn State Health. J.I.W. is supported by the Laurence M. Demers Career Development Professorship in Pathology and Medicine, Pennsylvania State University. J.D.R. is supported by the Ken and Bonnie Shockey Fund for Urologic Research at Penn State Health. J.M. is funded by a SEED Award from the HOPE Foundation, the Department of Defense (W81XWH-18-0257), and the VHA (BX003692-01).

## Author contributions
Conceptualization, N.M., F.X.R., S.V.L, F.P., P.L., and L.D.; methodology, S.V.L, F.P., P.L., A.T., C.S.G., K.B.D., T.S., I.N., E.C., M.S., A.R., N.B., L.M.S., T.S. (Pathology, Aarhus University Hospital), N.M., F.X.R., and L.D.; formal analysis, S.V.L, F.P., P.L., A.T., C.S.G., and M.S.; investigation, S.V.L., F.P., P.L., A.T., C.S.G., M.S., N.M., F.X.R., and L.D.; resources, K.B.D., J.B.J., G.G.H., A.C.P., V.W., M.G., M.H., G.S., M.H. (Lund University), K.M., R.N., B.J., X.L., D.D., D.W., A.G, R.A., C.D.H., J.D.R, J.I.W., U.S., D.S., K.E.M. van K., T.M., J.J.M., D.J.D., R.T.B., M.K., T.S. (University of Belgrade), A.H., E.Z., P.M., N.M., F.X.R., and L.D.; data curation, S.V.L, F.P., and P.L.; writing—original draft, S.V.L., F.P., P.L., and L.D.; writing—review and editing, all authors; supervision, L.D.; project administration, L.D.; funding acquisition, S.V.L., F.P., J.D.R., J.I.W., D.J.D., N.M., F.X.R., and L.D.

## Competing interests
L.D. has sponsored research agreements with C2i-genomics, Natera, AstraZeneca and Ferring, and has an advisory/consulting role at Ferring. J.B.J. has sponsored research agreements with Medac, Photocure ASA, Cephaid, Nucleix, Astellas and Ferring, and has an advisory board role at Olympus Europe, Cephaid and Ferring. J.D.R. is involved in a sponsored scientific study or trial with Pacific Edge Biotechnologies, MDxHealth and Urogen Pharma, is a consultant for Urogen Pharma, and has an investment interest in American Kidney Stone Management. J.J.M. is a consultant for Ferring, AstraZeneca, Janssen and participated in advisory boards for Foundation Medicine and Nucleix. R.T.B. has contributed to advisory boards for Olympus Medical Systems & Janssen, and undertakes research funded by UroGen Pharma and QED Therapeutics. The following authors declare no competing interests: S.V.L., F.P., P.L., A.T., C.S.G., K.B.D., T.S., I.N., E.C., M.S., N.J.B., L.M., G.G.H., A.C.P., V.W., M.O.G., M.H., G.S., M.H., T.S., K.M., A.R., R.M., B.J., X.L., D.D., D.G.W., A.G., C.D.H., J.I.W., U.S., D.S., K.E.M.K., T.M., D.J.D., M.A.K., T.S., A.H., E.C.Z., P.U.M., N.M., and F.X.R.

## Additional information

[1]Department of Molecular Medicine, Aarhus University Hospital, Aarhus N, Denmark. [2]Department of Clinical Medicine, Aarhus University, Aarhus, Denmark. [3]Cartes d'Identité des Tumeurs (CIT) Program, Ligue Nationale Contre le Cancer, Paris, France. [4]Oncologie Moleculaire, UMR144, Institut Curie, Paris, France. [5]Department of Urology, Aarhus University Hospital, Aarhus N, Denmark. [6]Department of Urology, Herlev hospital, Copenhagen University, Copenhagen, Denmark. [7]Department of Pathology, Aalborg University Hospital, Aalborg, Denmark. [8]Institute of Pathology, University Hospital Erlangen, Friedrich-Alexander-University Erlangen-Nuremberg, Erlangen, Germany. [9]Department of Urology, Jena University Hospital, Jena, Germany. [10]Department of Urology, Malteser Hospital St. Josephshospital, Krefeld Uerdingen, Krefeld, Germany. [11]Division of Urological Research, Department of Translational Medicine, Lund University, Skåne University Hospital, Malmö, Sweden. [12]Division of Oncology and Pathology, Department of Clinical Sciences, Lund University, Lund, Sweden. [13]Department of Pathology, Aarhus University Hospital, Aarhus N, Denmark. [14]Department of Urology, Technical University of Munich, Klinikum rechts der Isar, Munich, Germany. [15]Departments of Pathology, Urology, Biochemistry and Molecular Genetics, Northwestern University School of Medicine, Chicago, IL, USA. [16]Clinic of Urology, Clinical Centre of Serbia, Faculty of Medicine, University of Belgrade, Belgrade, Serbia. [17]Bladder Cancer Research Centre, Institute of Cancer and Genomic Sciences, College of Medicine and Dental Sciences, University of Birmingham, Birmingham, UK. [18]Leeds Institute of Medical Research at St James's, University of Leeds, Leeds, UK. [19]Department of Surgery, Division of Urology, Pennsylvania State University, Hershey, PA, USA. [20]Department of Pathology and Laboratory Medicine, Division of Urology, Department of Biochemistry and Molecular Biology, Pennsylvania State University, Hershey, PA, USA. [21]Department of Surgical Sciences, Uppsala University, Uppsala, Sweden. [22]Department of Urology and Pediatric Urology, University Hospital Erlangen, Friedrich-Alexander-University Erlangen-Nuremberg, Erlangen, Germany. [23]Department of Pathology, Erasmus MC Cancer Institute, Erasmus University Medical Center, Rotterdam, The Netherlands. [24]Department of Urology and Martini-Clinic, University Medical Center Hamburg-Eppendorf, Hamburg, Germany. [25]Institute of Medical and Clinical Biochemistry, Faculty of Medicine, University of Belgrade, Belgrade, Serbia. [26]Genetic and Molecular Epidemiology Group, Spanish National Cancer Research Center (CNIO), CIBERONC, Madrid, Spain. [27]Epithelial Carcinogenesis Group, Spanish National Cancer Research Center (CNIO), Madrid, Spain. [28]Departament de Ciències Experimentals i de la Salut, Universitat Pompeu Fabra, CIBERONC, Barcelona, Spain. [29]These authors contributed equally: Sia Viborg Lindskrog, Frederik Prip. ✉email: lars@clin.au.dk

