## [Peer Review File · Nature Communications]

REVIEWER COMMENTS

Reviewer #1 (Remarks to the Author): Expert in bladder cancer genomics and urology

The authors report an integrated "-omics" approach to non-muscle invasive bladder cancer. The cohort is a multi-institutional collaboration analyzing over 800 samples with subsets of samples being analyzed using RNA, DNA and protein methods. Overall the manuscript is well written and organized. The comprehensive analysis validates previously published data and with increased numbers analyzed starts to suggest how this information can become clinically relevant and used for therapeutic options. A few suggestions below:

-Figure 1 the authors used expression data in 535 patients with only 3 CIS patients to identify 4 clusters and used those to correlate PFS. we know that a small portion of pTa patients will progress. A sub analysis only with the pT1 and CIS patients would be more prudent as those are the patients most likely to progress. Being to better delineate the pT1/CIS, which may progress vs those that won't would be very clinically significant

-Similarly can RFS be analyzed for pTa sub-group

-the authors used their transcriptomic and genomic findings to try and improve on know risk stratifies. They used the EORTC, but would be interesting to see if AUA/SUO or CUETO models would have improvement as them may be better then EORTC.

John Sfakianos

Reviewer #2 (Remarks to the Author): Expert in bioinformatics and multi-omics analysis

Lindskrog et al. performed an integrative analysis of a very large number (n=862) of non-muscle-invasive bladder cancers (NMIBC), including bulk RNASeq (n=535), CNV/LOH by SNP array (n=473), spatial proteomics (n=167). From gene expression patterns, all NMIBC cases were classified into four distinct subgroups. The previous high-risk class 2 was found to contain two subgroups associated with different genes: class 2a with cell cycle & differentiation genes, and 2b with cancer stem cells and EMT genes, also with distinct clinical outcomes. This four-class stratification was further supported by additional evidence from the analyses of regulons, chromatin remodeling genes, and methylation sites. CNV/LOH analyses found chromosomal instability was associated with disease aggressiveness, which is interesting. Overall, this is a very large multi-omics study that provides novel insights into the disease heterogeneity of NMIBC. The findings are scientific interesting and clinically relevant. However, there are a few technical questions that could be better addressed.

Major:

1. A major technical issue is the approach of somatic mutation calling. Typically, the most common method for characterizing somatic point mutations (SNVs/Indels) is by whole-exome sequencing (WES) or whole-genome sequencing of paired tumor and germline samples. In this study, however, SNVs were mostly identified from RNASeq data, which also appeared to be unpaired without

germline. There are multiple potential problems with this type of approach: 1) unable to completely exclude germline SNPs. In the validation analysis by WES of 95 mutations, even after the authors performed extensive germline filtering, a high fraction (11/95) of remaining mutations still turned out to be germline SNPs. Related to this, the currently claimed 87% validation rate may not be accurate, as the germline SNPs should not be included with somatic mutations when calculating the validation rate. Further, germline SNPs should not be included in the calculation of mutation burden/signature. 2) in comparison with WES, it was unclear what % of mutations detected by WES were missed by the RNASeq-derived analysis. This is important to know because as the authors acknowledged, RNASeq can only detect mutations that are currently expressed (i.e. may miss the mutations expressed at different stages of tumor development), and may have poorer performance towards the mutations in genes at low expression levels. 3) for the “RNA-specific” or “Uncertain” mutations, it was not clear what mechanisms they represent. Are they present in the WES at low levels? And there is a lack of description of the method for confirming that these mutations were not present in the WES data. One thing they can probably do is to check the number of mutant reads and coverage of these RNA-specific mutations in the WES data. That would help determine if the absence of mutations were caused by technical issues such as low coverage, or false-negative by the caller. 4) it seems the current analysis did not consider Indels, which are common in some cancer genes such as TP53. If this is due to the technical difficulty of detecting Indels from RNASeq, then that should be discussed. The cited reference #44 to support RNA-derived mutation detection as “high-precision”, but it was not immediately clear how this conclusion was drawn. Overall, the current RNA-derived mutations could be further refined, and these limitations should be properly discussed.

Minor:

1. Only mutations of high and moderate functional effects were reported, which probably is understandable for identifying the driver mutations. For the calculation of the overall mutation burden, it was not clear if all somatic mutations were included.
2. For the CNV analysis, it is completely optional but might be of interest to also summarize genes affected by small focal CNVs and their association with different subtypes.
3. In figure 3a, it seems the class 1 and 3 have almost identical overlapping with GC1/2/3, despite their difference in survival etc, which looks interesting. Is there any possible explanation for that, e.x. genome instability happened at an earlier stage?
4. Figure 3H: it looks there are some patients with both CNV loss and point mutation in TP53. Are these mutations homozygous then? If so, is there any difference between patients that are TP53 homo vs hetero mutated in Figure 3I?
5. In the mutational signature analysis, only mutations with AF >0.15 and <0.60 were included. What was the rationale for excluding high AF mutations? How about homozygous somatic mutations?
6. There is a typo in Figure 5C: “Reccurrene rate”.

Reviewer #3 (Remarks to the Author): Expert in immunology

This paper builds on the previous UROMOL transcriptional analysis published in Cancer Cell 2016 by adding more samples and reanalysing the old data. Moreover, new multi-omics data have been

added. Importantly, the previous Class 2 is now subdivided to a and b with different characteristics.

Specific comment:

Figure 4a: Please, extend the legend by giving information about the antibody used and the colour it gives. Anti-CD8 staining in the low infiltration panel is not very convincing.

Reviewer #4 (Remarks to the Author): Expert in immunogenomics

This is an excellent paper that significantly contributes to the understanding of NMIBC. The manuscript is well written, the figures are (mostly) clear and the conclusions are appropriately stated. The paper also generates relevant data for more in depth studies.

Of the techniques utilized, proteomic assessment and documentation requires additional effort prior to publication such that the community can better understand and replicate the results.

Below are recommended edits:

1. Results/Delineation of transcriptomic classes in NMIBC:

Please further comment on class 3 for the reader as it establishes an early grounding for comparisons.

2. Integration of genomic alterations and transcriptomic classes: please remove "suggesting that these tumors present a high level of neoantigens" unless you have data to reference.

3. Spatial proteomics analysis: please revisit this section and describe with more detail and improved clarity:

- How are you defining "high class I"? what is your normalization modality?
- Please define your Z-score calculation.
- Fig. 4a is a poor representation of high vs. low in that it does not appear to represent the extent of difference in Fig. 4b. Please resolve as it suggests the data in 4b is not scaled appropriately. CTL and T-helper staining appears very weak suggesting technical problems in staining or acquisition thereof. Please specify in the legend what the colors refer to (what is yellow in line vs. globular yellow in image?).
- Please include single panel mplex images showing multiplex staining equivalency with DAB and adequate stripping of antibodies between cycles such that residual antibody is not causing signal in the next cycle of the tyramide assay (suppl). More details of the assay would be helpful as well (retrieval times, stripping times) which you may be able to reference.
- Please include more details for the DAB assays in this paper (a table of the retrieval time/conditions, antibodies, dilutions, incubation time etc similar to the mplex would be more complete than partial data listed in the methods).
- The use of overlays for PDL1 and CK is not ideal as the myeloid or TAM cells expression of PDL1 within or adjacent to tumor can be mis-attributed to tumor and vice versa. This should ideally be a

separate multiplex panel or caveats to interpretation included in the manuscript.

- There seems to be little to no mention of TMA core replicate statistics in the manuscript or of core-to-core variability in stroma or tumor region areas analyzed or variation of the total area of tumor and stroma regions analyzed between subgroups. Please further clarify.
- Please further clarify how the multiplex images are analyzed to derive single cell data (which version of software and modules are being used)?

Reviewers' comments

Reviewer #1 (Remarks to the Author): Expert in bladder cancer genomics and urology
The authors report an integrated "-omics" approach to non-muscle invasive bladder cancer. The cohort is a multi-institutional collaboration analyzing over 800 samples with subsets of samples being analyzed using RNA, DNA and protein methods. Overall the manuscript is well written and organized. The comprehensive analysis validates previously published data and with increased numbers analyzed starts to suggest how this information can become clinically relevant and used for therapeutic options. A few suggestions below:

Author response: We thank the Reviewer for the positive comments.

-Figure 1 the authors used expression data in 535 patients with only 3 CIS patients to identify 4 clusters and used those to correlate PFS. We know that a small portion of pTa patients will progress. A sub analysis only with the pT1 and CIS patients would be more prudent as those are the patients most likely to progress. Being to better delineate the pT1/CIS, which may progress vs those that won't would be very clinically significant

Author response: As pointed out by the reviewer, patients with T1 tumors have a higher progression rate than patients with Ta tumors. In our cohort, we see that 7.4% (29/393) of the patients with Ta tumors progress, whilst 26.1% (36/138) of the patients with T1/CIS tumors progress. However, even though most progression events are restricted to T1 tumors, some Ta tumors show a molecular profile associated with high progression risk. For example, transcriptomic class 2a includes some Ta tumors (Fig. 1d), and nearly half of the tumors in the genomic class 3 with high chromosomal instability are Ta tumors (Fig. 2d). We agree with the reviewer that it is highly important to delineate which T1/CIS tumors may progress, but a clinically relevant classifier should be able to identify most or all high risk cases - especially as there is a large variation between pathologists in the evaluation of tumor stage and grade.

However, the proposed sub-analysis may represent a useful way of eliminating interference resulting from stage and grade differences. Hence, we have now undertaken a sub-analysis of only T1 high grade (HG) tumors to generate data on a pathologically homogeneous tumor series, and this is now included as a new sub-chapter in the Results section: "Transcriptomic subtypes in pathologically homogeneous tumors" and in Supplementary Fig. 3. We conclude: "*The sub-analysis of pathologically homogeneous tumors demonstrates that the UROMOL2020 classes are not mainly driven by differences in histological and morphological features. The T1HG subtypes overlap partially with previously reported biological subtypes and signatures of aggressiveness; however, the increase in biological granularity is not directly translated into better prediction of outcome, since several progression events are missed using the T1HG classifier (Ta*

progression sensitivity: T1HG-1+3 subtype, 24% (7/29); UROMOL2020 class 2a+2b, 79% (23/29). T1 progression sensitivity: T1HG-1+3 subtype, 69% (25/36); UROMOL2020 class 2a+2b, 89% (32/36)”).

- Similarly can RFS be analyzed for pTa sub-group

Author response: In order to perform a sub-analysis of a pathologically homogeneous group of tumors, we have only focused on Ta low grade tumors. This is also included in the Results section under “Transcriptomic subtypes in pathologically homogeneous tumors” and in Supplementary Fig. 2. We write: “*Analysis of Ta low grade tumors by unsupervised consensus clustering of gene-based expression values restricted to the 2000 genes with highest variation identified four subtypes significantly overlapping with the UROMOL2020 classes ($p=4.4 \times 10^{-69}$; chi-square test; **Supplementary Fig. 2a**). The Ta low grade subtypes were, however, not significantly associated with RFS (**Supplementary Fig. 2b**)*”. Consequently, the analysis of pathologically homogeneous tumors supports the major UROMOL classes identified.

We would like to thank the reviewer for the suggestion to include these sub-analyses, as it strengthens the understanding of tumor biology and underlines the importance of the major classes.

- The authors used their transcriptomic and genomic findings to try and improve on known risk stratifiers. They used the EORTC, but would be interesting to see if AUA/SUO or CUETO models would have improvement as they may be better than EORTC.

Author response: We have now included comparisons to EAU risk scores - we also tried AUA, but as the difference between EAU and AUA is negligible in this work, we continued with EAU because of the European origin of the UROMOL consortium. In Supplementary Table 2, we compare our genomic and transcriptomic classes to both EORTC and EAU. Overall, the EORTC score is a stronger predictor than the EAU risk model, when stratified for molecular features (EAU low and intermediate risk were combined as no significant difference in progression-free survival was found in a univariate Cox regression analysis: intermediate vs. low, HR=3.00 (95%CI: 0.68-13.24), $p=0.15$). We also included the comparison to EAU risk scores in Supplementary Fig. 8b+d, where we observe that the AUC for EORTC is higher than for EAU. This is now discussed in the manuscript where we write: “*Logistic regression models including continuous variables (EORTC, genome altered and 12-gene progression score), EAU risk scores and T1HG subtypes showed no increased predictive value (**Supplementary Fig. 8a-d**)*.”

Reviewer #2 (Remarks to the Author): Expert in bioinformatics and multi-omics analysis

Lindskrog et al. performed an integrative analysis of a very large number ($n=862$) of non-muscle-invasive bladder cancers (NMIBC), including bulk RNASeq ($n=535$), CNV/LOH by SNP array ($n=473$), spatial proteomics ($n=167$). From gene expression patterns, all

NMIBC cases were classified into four distinct subgroups. The previous high-risk class 2 was found to contain two subgroups associated with different genes: class 2a with cell cycle & differentiation genes, and 2b with cancer stem cells and EMT genes, also with distinct clinical outcomes. This four-class stratification was further supported by additional evidence from the analyses of regulons, chromatin remodeling genes, and methylation sites. CNV/LOH analyses found chromosomal instability was associated with disease aggressiveness, which is interesting. Overall, this is a very large multi-omics study that provides novel insights into the disease heterogeneity of NMIBC. The findings are scientific interesting and clinically relevant. However, there are a few technical questions that could be better addressed.

Author response: We thank the Reviewer for the positive comments.

Major:

1. A major technical issue is the approach of somatic mutation calling. Typically, the most common method for characterizing somatic point mutations (SNVs/Indels) is by whole-exome sequencing (WES) or whole-genome sequencing of paired tumor and germline samples. In this study, however, SNVs were mostly identified from RNASeq data, which also appeared to be unpaired without germline. There are multiple potential problems with this type of approach:

1) unable to completely exclude germline SNPs. In the validation analysis by WES of 95 mutations, even after the authors performed extensive germline filtering, a high fraction (11/95) of remaining mutations still turned out to be germline SNPs. Related to this, the currently claimed 87% validation rate may not be accurate, as the germline SNPs should not be included with somatic mutations when calculating the validation rate. Further, germline SNPs should not be included in the calculation of mutation burden/signature.

Author response: Overall, we completely agree that calling somatic variants based on RNA-Seq data is not optimal when compared to DNA-based methods like WES or WGS. We apply several filtering steps to avoid including too many germline SNPs and, in particular, the size of the dataset makes it also possible for us to filter across samples. We acknowledge that there are issues with this approach and discuss this in the manuscript. Furthermore, it is important to stress that we mostly focus on already known mutated genes in bladder cancer (Fig. 3e) and known mutational signatures.

We agree with the reviewer that it is not possible to completely exclude germline SNPs but we have included many filters to remove most of them:

- we remove any mutations annotated with a rs ID
- we remove any mutations common to more than 10 samples in our 535 patients cohorts except for a few hotspots positions in FGFR3 and PIK3CA.

In addition, we only reported genes that are known in bladder cancer in Figure 3, and genes with significantly different mutation patterns across classes in Supplementary Fig. 5. We do not expect to see those genes to be differentially affected between the classes by germline SNPs.

Notwithstanding, we have now included additional validity analyses by investigating all of the mutations (hotspot mutations were not included) called from the RNA-sequencing

data in 38 samples where we have both DNA and RNA sequencing data available (n = 11,016 mutations). We observed that:

- 32.5% were observed with a frequency above 2% only in the tumor DNA
- 0.8% only in the germline DNA (with very low VAF but above 2%)
- 20.6% in both tumor and germline (VAF > 2%; probably germline SNPs)
- 46% were unique to RNA (VAF < 2% in both germline and tumor DNA; RNA specific events).

Importantly, when we restrain our analysis to the genes shown in Fig. 3e or to Supplementary Fig. 5b genes, the proportion of true somatic SNVs rose to 79.8% for Fig. 3e (74 / 93 mutations) and 68.8% for Supplementary Fig. 5 (190 / 280 mutations). These results are now displayed in Fig. 3f in the manuscript and we specifically write: *“We compared RNA-Seq and whole exome sequencing (WES) of tumors and germline for 38 patients, and found that the filtering approach applied per sample and across samples enriched significantly for somatic SNVs in our presented gene lists (Fig. 3f).”* Consequently, the process of filtering for frequently mutated and significantly differentially mutated genes enrich for somatic variants. We have listed these numbers in Fig. 3f as indicated, and have now emphasized that only the frequency of somatic variants is considered as validated.

Concerning the mutational burden, we also agree that germline SNPs should not be included in the calculation; but since it is not possible to remove them in the absence of germline DNA sequencing for all the samples, then we cannot achieve this. However, we have now compared the number of mutations called from RNA (using the 791 genes used for calculating the RNA-derived tumor mutational burden) and the number of mutations called from DNA (with VAF > 10%), and have obtained a correlation of 0.74. We feel that these data demonstrate that the RNA-based TMB we present represents a good proxy for DNA-based TMB. A supplementary figure (see below) that shows the correlation between the two measures is now added to Supplementary Fig. 5.

For the mutational signatures, we restricted our analysis to the APOBEC signature by looking at the proportion of C to G/T mutations in a TCW context in both RNA and DNA from the 38 matched samples. Here we obtained a correlation of 0.75, demonstrating that APOBEC contribution can be inferred from RNA mutation calling despite the presence of germline SNPs and RNA-specific mutations. The figure has also been added to Supplementary Fig. 5.

2) in comparison with WES, it was unclear what % of mutations detected by WES were missed by the RNASeq-derived analysis. This is important to know because as the authors acknowledged, RNASeq can only detect mutations that are currently expressed (i.e. may miss the mutations expressed at different stages of tumor development), and may have poorer performance towards the mutations in genes at low expression levels.

Author response: We thank the Reviewer for the comment and have now added an analysis addressing these concerns. First, we looked at the proportion of WES calls with an allele frequency above 10% that are called in the RNA analysis or observed with 1, 2 or 3 reads. We restricted the analysis to positions with at least 10 reads.

This plot shows that more than 81% of the WES calls are called in the RNA analysis when the VAF in the DNA is above 20% (black). It also shows that we can observe one read (red) for only 96% of them, 2 reads (green) for about 95% and 3 reads (blue) for about 94% of them.

To address the problem of low expression, we considered all of the WES calls with a frequency above 20% and calculated the proportion of calls in the RNA analysis or the number of reads showing the alternate allele as a function of the number of reads at the given position.

This shows that there is an increase in accuracy for calling a WES mutation from 1 read to 20 reads followed by a stable and good level of recall. As a Reviewer suggested, this means that we have a poorer performance when looking at genes with lower expression which can be improved by deeper sequencing.

Finally, this plot shows that in order to get a near optimal performance, only RNA regions with more than 20 reads should be taken into account. From ~81% at 10 reads or more, we to achieve ~86% at 20 reads or more. However, with our filter, we cannot get above 90% (at 100 reads or more), probably because of allelic specific expression.

The plots presented here regarding the recall of DNA muts in RNA, mutational burden and APOBEC-related mutations have been added to Supplementary Fig. 5 and we now write: “*We compared RNA-Seq and whole exome sequencing (WES) of tumors and germline for 38 patients, and found that the filtering approach applied per sample and across samples enriched significantly for somatic SNVs in our presented gene lists (Fig. 3f). Additional comparative analysis of mutations observed in DNA documented a high correlation between observation in DNA and RNA (Supplementary Fig. 5c-e), suggesting that potential included germline variants have limited impact on subsequent analyses.*”

3) for the “RNA-specific” or “Uncertain” mutations, it was not clear what mechanisms they represent. Are they present in the WES at low levels? And there is a lack of description of the method for confirming that these mutations were not present in the WES data. One thing they can probably do is to check the number of mutant reads and coverage of these RNA-specific mutations in the WES data. That would help determine if the absence of mutations were caused by technical issues such as low coverage, or false-negative by the caller.

Author response: We thank the Reviewer for the comment. We have now changed Fig. 3f and added a better description of the method. To summarize, in order to validate RNA calls in the DNA, we required a minimum of 20 reads in both the tumor and the germline DNA sequencing data. A mutation was denoted as RNA-specific if, and only if, the VAF of the alternate allele in the tumor and in the germline DNA sequencing data was below 2%.

4) it seems the current analysis did not consider Indels, which are common in some cancer genes such as TP53. If this is due to the technical difficulty of detecting Indels from RNASeq, then that should be discussed. The cited reference #44 to support RNA-derived mutation detection as “high-precision”, but it was not immediately clear how this conclusion was drawn. Overall, the current RNA-derived mutations could be further refined, and these limitations should be properly discussed.

Author response: Indels are a bit more difficult to call in the RNA and we decided not to include it in this work. Different technical issues are arising when considering indels. First, mid-size insertions and deletions in RNA-seq short reads can be seen by the aligner as splicing events. On the other hand, some non-canonical splicing events could be falsely considered as indels in the RNA-seq. Second, there are no tools to really separate somatic and germline indels in the same proportion as for SNVs. Finally, indels may have a strong impact in the RNA-sequence deriving from it making it very difficult to validate. All those reasons made us decide not to look at indels calling from RNA-seq data. We have now underlined this in the method section where we write: “*Single base mutations*

were called from the RNA-seq data using the GATK pipeline. Indels were not considered here due to technical issues that may arise from calling this from RNA-Seq data”.

Minor:

1. Only mutations of high and moderate functional effects were reported, which probably is understandable for identifying the driver mutations. For the calculation of the overall mutation burden, it was not clear if all somatic mutations were included.

Author response: Please see response to major point 1.

2. For the CNV analysis, it is completely optional but might be of interest to also summarize genes affected by small focal CNVs and their association with different subtypes.

Author response: Copy number alterations in specific known bladder cancer genes are included in Fig. 3e.

3. In figure 3a, it seems the class 1 and 3 have almost identical overlapping with GC1/2/3, despite their difference in survival etc, which looks interesting. Is there any possible explanation for that, e.x. genome instability happened at an earlier stage?

Author response: Class 1 and 3 tumors have very similar outcomes (Fig. 1b, Supplementary Table 2), but the biology is different, as highlighted in the Results section and in the overview Fig. 5c. We have not observed any differences in genomic instability between the groups.

4. Figure 3H: it looks there are some patients with both CNV loss and point mutation in TP53. Are these mutations homozygous then? If so, is there any difference between patients that are TP53 homo vs hetero mutated in Figure 3I?

Author response: Yes, the majority of *TP53* mutations in tumors with both copy number (CN) change and point mutation in *TP53* are in fact homozygous (mean VAF for samples with CN change = 0.89, mean VAF for samples with no CN change = 0.65), and there is a significant correlation between the RNA-derived VAF and the amount of genome altered (see Figure below). We now comment on this in the manuscript and specifically write: “*TP53* was affected by both copy number change and point mutation in 17 tumors (Fig. 3h), and the majority of these mutations were homozygous (mean variant allele frequency was 0.89 in tumors with copy number change and 0.65 in tumors without). Furthermore, we found a positive correlation between *TP53* variant allele frequency and genomic changes ($R=0.44$, $p\text{-value}=0.027$; Pearson’s correlation)”

5. In the mutational signature analysis, only mutations with AF >0.15 and <0.60 were included. What was the rationale for excluding high AF mutations? How about homozygous somatic mutations?

Author response: We believe that when including all synonymous mutations in the mutational signature analysis, then the probability of a high VAF mutation to be a germline SNP is much higher than for it to be a true homozygous somatic mutation. Therefore, we discarded them to minimise the risk of inclusion of germline SNPs. Our APOBEC sub-analysis shows a very good correlation between RNA and DNA APOBEC contribution (see answer to major point 1).

6. There is a typo in Figure 5C: “Recurrene rate”.

Author response: Thank you - this is now corrected.

Reviewer #3 (Remarks to the Author): Expert in immunology

This paper builds on the previous UROMOL transcriptional analysis published in Cancer Cell 2016 by adding more samples and reanalysing the old data. Moreover, new multi-

omics data have been added. Importantly, the previous Class 2 is now subdivided to a and b with different characteristics.

Author response: We thank the Reviewer for the comment.

Specific comment:

Figure 4a: Please, extend the legend by giving information about the antibody used and the colour it gives. Anti-CD8 staining in the low infiltration panel is not very convincing.

Author response: We have extended the legend to include information on the antibodies and fluorophores used. Furthermore, a full list of antibodies and fluorophores can be found in Supplementary Data 1. We have now fully replaced and redesigned the illustrations in Fig. 4a with more representative examples of tumors with high- and low infiltration, respectively. In addition, we have added a scale-bar to ease interpretation.

Reviewer #4 (Remarks to the Author): Expert in immunogenomics

This is an excellent paper that significantly contributes to the understanding of NMIBC. The manuscript is well written, the figures are (mostly) clear and the conclusions are appropriately stated. The paper also generates relevant data for more in depth studies.

Author response: We thank the Reviewer for the positive comments.

Of the techniques utilized, proteomic assessment and documentation requires additional effort prior to publication such that the community can better understand and replicate the results.

Below are recommended edits:

1. Results/Delineation of transcriptomic classes in NMIBC:

Please further comment on class 3 for the reader as it establishes an early grounding for comparisons.

Author response: We thank the reviewer for this suggestion. In the “Delineation of transcriptomic classes in NMIBC” section, we mention that class 3 is associated with a higher expression of early cell-cycle genes, differs from class 1 by having high AR and GATA3 regulon activity and less methylated gene promoters and shows significantly lower immune infiltration compared to all other classes. We have now added the following sentence: “*Furthermore, class 3 tumors were characterized by high expression of FGFR3-coexpressed genes and a depleted immune contexture (Fig. 1e-f), as previously demonstrated in MIBC and upper tract urothelial carcinoma (Sweis et al., Cancer Immunol Res., PMID: 27197067; Robinson et al., Nat Commun. 2019, PMID: 31278255)*”. The biology of class 3 tumors is also included in the discussion (fourth paragraph), and potentially druggable pathways in class 3 are suggested (fifth paragraph). Furthermore, a summarised overview of class 3 tumors is provided in Fig. 5c.

2. Integration of genomic alterations and transcriptomic classes: please remove "suggesting that these tumors present a high level of neoantigens" unless you have data to reference.

Author response: We agree with the Reviewer that this statement requires data to reference and it is now removed.

3. Spatial proteomics analysis: please revisit this section and describe with more detail and improved clarity:

- How are you defining "high class I"? what is your normalization modality?

Author response: We are not entirely clear as to what the reviewer is referring to with regard to "high class I", and so apologize if we do not fully address this question. However, in the section regarding the WISP analysis, we describe high class 1 weights, which are defined in the Methods section relating to that component of the analysis. We have now included an additional sentence in the Results section for further clarification: "*WISP calculates pure population centroid profiles from the RNA-Seq data and estimates class weights for each sample based on the centroids (hence, each sample is weighted between all four transcriptomic classes; for details see **Methods**).*"

- Please define your Z-score calculation.

Author response: We have now included the calculation of the z-score in the legend to

Fig. 4b: $Z \text{ score} = \frac{\text{value} - \text{mean}}{\text{standard deviation}}$

- Fig. 4a is a poor representation of high vs. low in that it does not appear to represent the extent of difference in Fig. 4b. Please resolve as it suggests the data in 4b is not scaled appropriately. CTL and Thelper staining appears very weak suggesting technical problems in staining or acquisition thereof. Please specify in the legend what the colors refer to (what is yellow in line vs. globular yellow in image?).

Author response: We thank the reviewer for highlighting that the weak staining intensity in Fig. 4a could be misinterpreted as technical issues. However, in this case the weak colouring appeared after exporting the scanned image from the Visiopharm software. Hence, as also mentioned in the response to Reviewer 3, we have now replaced and redesigned the illustrations in Fig. 4a with more representative high-quality images of tumors with high- and low immune cell infiltration, respectively. We have extended the legend to include information on antibodies, fluorophores, and a reference to the dashed line separating the tumor parenchyma from the tumor stroma. Lastly, we have added a scale-bar to ease interpretation.

- Please include single panel mplex images showing multiplex staining equivalency with DAB and adequate stripping of antibodies between cycles such that residual antibody is not causing signal in the next cycle of the tyramide assay (suppl). More details of the

assay would be helpful as well (retrieval times, stripping times) which you may be able to reference.

Author response: We would like to refer the Reviewer to Supplementary Fig. 7. The figure includes single-plex images of the different fluorophores together with DAPI. We agree with the reviewer that the Methods section “Proteomics” should include more details and so to address this, we have rewritten the section and added a relevant reference. “Proteomics” now includes an additional subsection “Immunofluorescence, immunohistochemistry and imaging”, incorporating all of the specific details relating to reagents, retrieval times, stripping times, ect. for fluorescence and brightfield detection. Detailed information on the antibody assays (clone, company, species, dilution, incubation, RRID and fluorophore) is listed in Supplementary Data 1.

- Please include more details for the DAB assays in this paper (a table of the retrieval time/conditions, antibodies, dilutions, incubation time etc similar to the mplex would be more complete than partial data listed in the methods).

Author response: Details for the DAB assays is accessible in Supplementary Data 1. This has now also been clarified in the Methods section.

- The use of overlays for PDL1 and CK is not ideal as the myeloid or TAM cells expression of PDL1 within or adjacent to tumor can be mis-attributed to tumor and vice versa. This should ideally be a separate multiplex panel or caveats to interpretation included in the manuscript.

Author response: We agree with the Reviewer that it is not possible for us to distinguish between PD-L1 positive immune cells and PD-L1 positive carcinoma cells. We did not adequately specify this in the paragraph regarding PD-L1 expression. We have now clarified this in the revised manuscript, and we now operate with overall PD-L1 expression in the different tumor regions (stroma and parenchyma). We thank the Reviewer for highlighting this, and for allowing us to clarify the manuscript.

- There seems to be little to no mention of TMA core replicate statistics in the manuscript or of core-to-core variability in stroma or tumor region areas analyzed or variation of the total area of tumor and stroma regions analyzed between subgroups. Please further clarify.

Author response: We have now included core-to-core correlation statistics to the Methods section. We generally observe a strong correlation between the TMA tissue cores (see figure below). The correlation is strongest for immune cells in the tumor parenchyma and less for immune cells in the tumor stroma; however, still significant. In the manuscript we have mainly focused on infiltrating immune cells present in the tumor parenchyma.

% of cells classified as immune cells. The distribution of immune cells is shown on the diagonal (histograms). On the bottom of the diagonal : Scatter plots with a fitted line. On the top of the diagonal : the value of the pearson correlation plus the significance level as stars. Each significance level is associated to a symbol : p-values(0.001, 0.01, 0.05, 0.1) => symbols(***, **, *, .)

The observed variation, especially in the tumor stroma, could be explained by the modest number of immune cells per tissue core (see figure below). We have therefore concluded that the per core number of immune cells was too limited to study intra-tumor heterogeneity. The low number of immune cells per core highlights the need for 3 cores per patient. The agreement between TMA sections and whole tissue sections has been studied in melanoma tumors for CD8 and CD163 and ranged from 83% to 96% (*Jensen et al., Tumor and inflammation markers in melanoma using tissue microarrays: a validation study. Melanoma Res., 2011*).

- Please further clarify how the multiplex images are analyzed to derive single cell data (which version of software and modules are being used)?

Author response: We have now included a detailed Methods section regarding digital pathology that specifies the software used (Visiopharm) and which modules we have utilized (Tissue array, Tissue align and Tissue author).

REVIEWERS' COMMENTS

Reviewer #2 (Remarks to the Author):

All my comments have been addressed.

Reviewer #5 (Remarks to the Author):

The authors applied a multi-omics approach in combination with taking advantage of a multi-center study to understand molecular networks in non-muscle-invasive bladder cancer to identify potential biomarkers of clinical outcome. The study is well-designed and represented in a clear and comprehensive way.

The authors carefully addressed all major concerns of reviewer 4.